# Escaping the Homophily Trap: A Threshold-free Graph Outlier Detection Framework via Clustering-guided Edge Reweighting

**Yunhe Zhang**[1*]  **Jinyu Cai**[2*]  **Qi Hao**[1]  **Pengyang Wang**[1†]  **See-Kiong Ng**[2]

[1]The State Key Laboratory of Internet of Things for Smart City, Department of Computer and Information Science, University of Macau, China

[2]Institute of Data Science, National University of Singapore, Singapore

{zhangyhannie,jinyucai1995}@gmail.com

asteriaqq.xin@connect.um.edu.mo

pywang@um.edu.mo  seekiong@nus.edu.sg

## Abstract

Graph outlier detection is a critical task for identifying rare, deviant patterns in graph-structured data. However, prevalent methods based on graph convolution are fundamentally challenged by the "Homophily Trap": the aggregation of features from neighboring nodes inadvertently contaminates the representations of normal nodes near anomalies, blurring their distinctions. To overcome this limitation, we propose a Clustering-guided Edge Reweighting framework for Graph Outlier Detection (CER-GOD), which jointly optimizes a self-discriminative masking spoiler and an adaptive clustering-based outlier detector. The masking spoiler learns to selectively weaken the influence of heterogeneous neighbors, preserving the discriminative power of node embeddings. This process is guided by the clustering detector, which generates pseudo-labels in an unsupervised manner, thereby eliminating the need for predefined anomaly thresholds. To ensure robust optimization and prevent class collapse—a failure mode exacerbated by the homophily trap—we introduce a diversity loss that stabilizes the clustering process. Our end-to-end framework demonstrates superior performance on multiple benchmark datasets, establishing a new state-of-the-art by effectively dismantling the homophily trap.

## 1 Introduction

Graph outlier detection, which aims to identify anomalous data (*e.g.*, nodes, subgraphs) deviating from dominant patterns, is a critical unsupervised learning task with significant real-world applications in areas like financial fraud detection (Kim et al., 2024; Wang et al., 2019; **?**), traffic monitoring (Wawrowski et al., 2023; Le et al., 2011; Zhou et al., 2009), and biological analysis (Zhou et al., 2025; Xu et al., 2024), *etc*. Over the past decades, a variety of detection strategies have emerged, achieving remarkable success, such as reconstruction-based measurements (Ding et al., 2019; Fan et al., 2020), contrastive learning based strategies (Liu et al., 2021b; Dillon et al., 2024), or statistical characteristic-based methods (Chen et al., 2020; Breunig et al., 2000).

Despite their diverse approaches, a foundational component in many state-of-the-art models is the graph convolutional (GC) operation, which learns node representations by aggregating information from local neighborhoods (Xu et al., 2019; Welling & Kipf, 2016; Sun et al., 2019). However, the effectiveness of GC operations is rooted in the principle of homophily—the assumption that nearby nodes are similar. This very principle creates a fundamental conflict in outlier detection. When normal and anomalous nodes are neighbors, the convolution process blurs their distinctions, a problem recently termed the "Homophily Trap" (He et al., 2024). This contamination of node embeddings obscures the discriminative features essential for identifying anomalies, thereby undermining the performance of existing detectors.

---

*These authors contributed equally.

†Corresponding author.

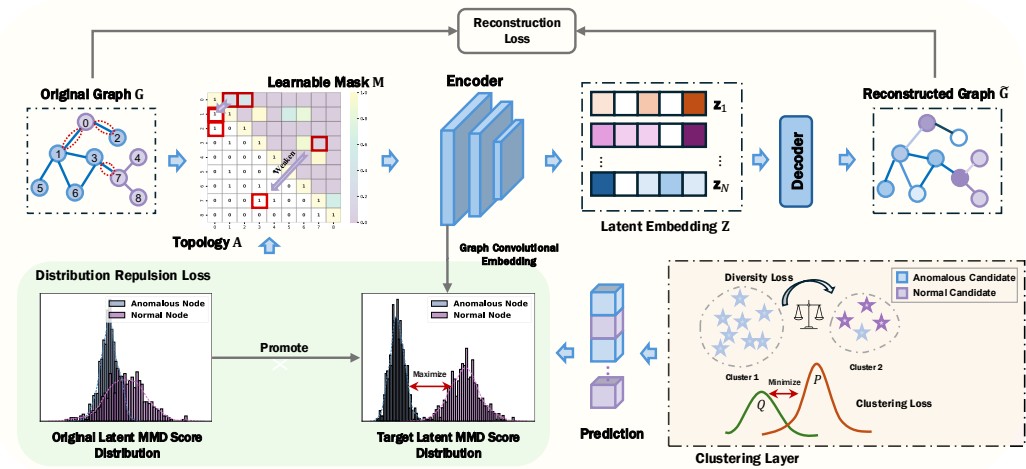

Figure 1: The architecture of the proposed CER-GOD framework for graph outlier detection. The model takes an input graph with its topology, applies a learnable mask to suppress noisy or irrelevant connections, and encodes the refined structure using graph convolutional layers. The latent embeddings are then used for graph reconstruction and clustering-based anomaly prediction. Based on these predictions, normal and anomalous candidate groups are generated and optimized through distribution repulsion loss. The framework is jointly optimized with three objectives: reconstruction loss, clustering loss (with a diversity regularization term), and a distribution repulsion loss.

To facilitate the understanding of the adverse effect of "Homophily Trap" in graph outlier detection, here we show empirical evidence in Figure 2. Formally, a 'normal node 1-hop away from anomalies' denotes a normal node with a direct edge (1-hop distance) to at least one anomalous node, while a 'normal node multi-hop away from anomalies' is a normal node at a distance of two or more hops from its nearest anomalous neighbor. The embeddings of neighboring normal nodes are noticeably altered after graph convolution, with minimal distinction between 1-hop normal and anomalous nodes. This illustrates that: **(1)** anomalous neighbors can weaken the discriminability of normal nodes; and **(2)** this contamination effect decreases with increasing path. For example, when normal nodes are surrounded by anomalous ones (or vice versa), the aggregation process can blur the distinction between them, thereby weakening the model's ability to detect outliers. This issue is especially severe

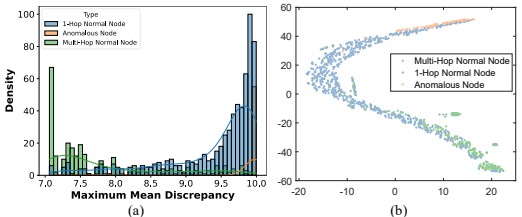

Figure 2: The histograms of maximum mean discrepancy distances and the t-SNE visualization between standard Gaussian distribution $\mathcal{N}(\mathbf{0}, \mathbf{I}_d)$ and three types of node embeddings (normal node multi-hop away from anomalies, normal node 1-hop away from anomalies, and anomalous nodes) for real-world anomaly on Email dataset. Note that the embeddings are obtained via a single-layer graph convolution operation.

when the nodes are closely connected, as anomalous neighbors exert stronger influence through short-range paths. In contrast, the contamination effect diminishes as the path length increases, suggesting that distant neighbors contribute less to the node's final representation.

To address this fundamental challenge, we propose a novel framework, termed **CER-GOD** (**C**lustering-guided **E**dge **R**eweighting for **G**raph **O**utlier **D**etection), which dismantles the homophily trap through the joint optimization of two synergistic components: 1) **Self-Discriminative Masking Spoiler**, and 2) **Clustering-based Outlier Detector**, where the architecture is shown in Figure 1. Specifically, the masking spoiler adaptively fine-tunes the edge weights of the original topology, thereby weakening or strengthening the degree of information aggregation in GC operations. As a result, the discriminability between the aggregated embeddings of normal and abnormal nodes is then enhanced. However, this becomes particularly challenging when node labels are unavailable. Thus,

we further introduce a clustering-based outlier detector, which eliminates the need for predefined thresholds when identifying anomalies. This enables the identification of candidate normal and anomalous nodes, which are then jointly optimized with the masking spoiler to further prevent the aggregation of heterogeneous node types.

Nevertheless, the "Homophily Trap" may potentially lead to class collapse (*i.e.*, all instances are clustered into a single group) due to incomplete or suboptimal optimization in the early learning process. To counter this, we further develop a diversity loss that is triggered when class collapse occurs and gradually reallocates a portion of samples from the dominant cluster to another cluster. This ensures the anomalous candidate group remains populated. Finally, we design a new anomalous score according to the probability confidence learned via the clustering layer. We demonstrate the superiority of the proposed method over state-of-the-art graph outlier baselines through comprehensive experiments on multiple benchmark datasets. The main contributions of this paper are summarized as follows:

- **A Novel Approach to Counter the Homophily Trap**: We provide a rigorous analysis of the "Homophily Trap" and introduce a self-discriminative masking spoiler that adaptively re-weights the graph topology to mitigate the contaminating influence of heterogeneous neighbors.

- **Threshold-Free Anomaly Detection**: We propose an adaptive clustering-based detector that generates pseudo-labels to guide the masking process in a fully unsupervised manner, eliminating the reliance on arbitrary, predefined thresholds for outlier identification.

- **Robust Optimization with Diversity Loss**: We introduce a diversity loss function that effectively prevents class collapse during clustering, ensuring the stability and reliability of the joint optimization framework.

## 2 METHODOLOGY

### 2.1 PRELIMINARY AND MOTIVATION

Given a graph $\mathcal{G} = \{\mathcal{V}, \mathcal{E}\}$, where $\mathcal{V} = \{v_1, v_2, \ldots, v_N\}$ denotes the set of nodes and $\mathcal{E}$ denotes the set of edges, each node is associated with an attribute vector $\mathbf{x} \in \mathbb{R}^d$, and the graph structure is represented by an adjacency matrix $\mathbf{A} \in \{0,1\}^{N \times N}$, where $\mathbf{A}_{ij} = 1$ if there exists an edge between node $v_i$ and node $v_j$. In graph outlier detection tasks, the objective is to learn a discriminative embedding in a latent space that effectively separates normal nodes from anomalous ones, which can be initially formulated as follows:

$$\max_f \ell(f(\mathcal{G}_{\text{normal}}), f(\mathcal{G}_{\text{abnormal}})), s(f(\mathcal{G})) = \begin{cases} 0, & s_i \leq \tau, \\ 1, & \text{otherwise}, \end{cases} \quad (1)$$

where $\ell(\cdot)$, $f(\cdot)$ and $s(\cdot)$ denote the distribution measurement, the graph representation learner, and the anomaly detector that assigns an anomaly score $s_i$ to each node, respectively. If $s_i$ exceeds the threshold $\tau$, the $i$-th node is classified as anomalous, and its label $y_i$ is set to 1.

To facilitate the learning of graph embeddings, an $L$-layer Graph Convolutional Network is utilized to learn the node representation at $l$-th layer:

$$\mathbf{z}_i^{(l)} = \text{Aggr}(\mathbf{z}_i^{(l-1)}, \mathbf{z}_j^{(l-1)} : j \in \mathcal{N}(i)), \quad (2)$$

where $\mathcal{N}(i)$ is the neighbor node set of the $i$-th node, $\text{Aggr}(\cdot)$ denotes the operation that updates the representation of a node by aggregating the information of its neighbors. $\mathbf{z}_i \in \mathbb{R}^k$ denotes the latent representation of node $v_i$, initialized as $\mathbf{z}_i^{(0)} = \mathbf{x}_i$. Although aggregating information along with the graph structure has achieved success in all kinds of fields, for the outlier detection task, it still brings several challenges:

- The aggregation operation along graph edges may propagate anomalous information from anomalous nodes to neighboring normal nodes, thereby contaminating their representations.

- According to the over-squashing phenomenon (Topping et al., 2022), as the shortest path distance between anomalous and normal nodes increases, the extent of information propagation (which is also known as the degree of contamination) diminishes.

Towards illustrating these challenges, let the shortest distance between node $i$ and $j$ be $r$, we use Jacobian matrix $\frac{\partial \mathbf{z}_j^{(r)}}{\partial \mathbf{x}_i}$ to quantify the influence of the node representation $\mathbf{z}_j^{(r)}$ to a specific input feature $\mathbf{x}_i$ in the node $i$.

**Proposition 1.** *If $|\nabla \sigma_l| \leq \alpha$ and $|\nabla Aggr_l| \leq \beta$ for $0 \leq l \leq r$, then the following holds*

$$|\frac{\partial \mathbf{z}_j^{(r+1)}}{\partial \mathbf{x}_i}| \leq (\alpha\beta)^{r+1}(\mathbf{A}^{r+1})_{ji}. \tag{3}$$

The proof of Proposition 1 can be found in Appendix A. Given the implications of it, we recognize that as the shortest path distance increases, the upper bound on the influence exerted by the $i$-th node's attributes on the target node embedding $\mathbf{z}_j^{(r)}$ progressively diminishes. It indicates that the sensitivity between them exhibits an exponentially decreasing trend with respect to their shortest path distance.

In the challenging unsupervised context of outlier detection, the "homophily trap" issue severely hinders the separation of normal nodes from nearby anomalies. Our approach directly confronts this by integrating a self-discriminative masking spoiler with an adaptive clustering-based detector. The masking spoiler encourages aggregation to perform primarily within intra-cluster samples and designs a penalty strategy to resist cluster collapse. Different from existing methods (please refer to Section 3), the proposed masking spoiler selectively weakens existing edges without altering the original message passing paths. This design preserves the structural integrity of the graph while encouraging the aggregated embeddings of the two clusters to become as discriminative as possible.

## 2.2 SELF-DISCRIMINATIVE MASKING SPOILER

We first present a mathematical formulation of the proposed candidate filter strategy. To begin with, a reconstruction-based Graph Auto-Encoder (GAE) is utilized to learn low-dimensional embeddings for each node:

$$\ell_r = \frac{1}{N} \sum_{i=1}^{N} (\|\hat{\mathbf{x}}_i - \mathbf{x}_i\|^2 + \|\hat{\mathbf{A}}_i - \mathbf{A}_i\|^2). \tag{4}$$

Here, $\hat{\mathbf{A}} = \text{sigmoid}(\mathbf{Z}^\top \mathbf{Z})$ denotes the reconstructed adjacency matrix, with $\mathbf{Z} = f_{\mathcal{W}}^{\text{enc}}(\mathbf{X}, \mathbf{A})$ representing the node embeddings obtained from the graph encoder function $f^{\text{enc}}(\cdot)$ parameterized by $\mathcal{W} = \{\mathbf{W}_l, \mathbf{b}_l\}_{l=1}^{L}$.

Then we introduce a learnable variable $\mathbf{M}$, and constrain the new topology $\tilde{\mathbf{A}}$ at each epoch based on the original topology $\mathbf{A}$ and $\mathbf{M}$:

$$\tilde{\mathbf{A}} = \tilde{\mathbf{M}} \odot \mathbf{A}, \quad \text{subject to} \quad \tilde{\mathbf{M}} \in [0, 1]^{N \times N}, \tag{5}$$

where $\odot$ denotes the Hadamard product and $\tilde{\mathbf{M}}$ is obtained through $\tilde{\mathbf{M}}_{ij} = \text{sigmoid}(\mathbf{M}_{ij})$. Following this, the normalization operation of the adjacency matrix $\tilde{\mathbf{A}} + \mathbf{I}_N$ is adopted to ensure that each node's ego-information is preserved. We define $\mathbf{I}_N$ as the identity matrix of shape $N \times N$.

The objective of this spoiler is to let the distributions of predicted normal nodes and anomalous nodes be as discriminative as possible. Here we first collect all predicted normal instances with label $y_i = 0$ into the normal candidate node set $\mathcal{D}_{\text{pos}}$, and the remaining instances into the anomalous candidate node set $\mathcal{D}_{\text{neg}}$. The maximum mean discrepancy (MMD) (Gretton et al., 2012) is then used to measure the aggregated distribution distance between the two groups. Based on this, we can define the distribution repulsion loss as follows:

$$\begin{aligned}
\text{MMD}^2[\mathcal{F}, \mathcal{D}_{\text{pos}}, \mathcal{D}_{\text{neg}}] =& \frac{1}{m(m-1)} \sum_{i=1}^{m} \sum_{j=1, j \neq i}^{m} \kappa(\mathbf{z}_i^{\text{pos}}, \mathbf{z}_j^{\text{pos}}) \\
&+ \frac{1}{n(n-1)} \sum_{i=1}^{n} \sum_{j=1, j \neq i}^{n} \kappa(\mathbf{z}_i^{\text{neg}}, \mathbf{z}_j^{\text{neg}}) - \frac{2}{mn} \sum_{i=1}^{m} \sum_{j=1}^{n} \kappa(\mathbf{z}_i^{\text{pos}}, \mathbf{z}_j^{\text{neg}}),
\end{aligned} \tag{6}$$

where $\kappa(\cdot, \cdot)$ represents a certain kernel function and $\mathcal{D} = \{\mathbf{z}_i^{(0)}\}_{i=1}^{N}$. Note that $\mathcal{D}$ can be taken as the graph convolutional outputs of an arbitrary layer when considering computational limits, though

this may reduce the optimization strength. Besides, we use the Gaussian kernel via the Chebyshev distance, which can be defined as:

$$\kappa_{\text{Chebyshev}}(x, y) = \exp\left(-\frac{d_{\text{Chebyshev}}(x, y)^2}{2\sigma^2}\right) = \exp\left(-\frac{(\max_i |x_i - y_i|)^2}{2\sigma^2}\right). \tag{7}$$

Since we apply it to a certain graph convolutional layer in high-dimensional space, focusing on the maximum difference in any single dimension is more effective, as it reduces sensitivity to noise in the remaining dimensions. Then we define the distribution repulsion loss as follows:

$$\ell_{\text{dr}} = -\text{MMD}^2(\mathcal{D}_{\text{pos}}, \mathcal{D}_{\text{neg}}). \tag{8}$$

Maximizing the separation between normal and anomalous points is straightforward in a supervised setting, where labels are available. However, in the absence of labels, achieving this goal becomes significantly more challenging, as obtaining reliable labels itself is a difficult task. To assist in optimizing the mask $\mathbf{M}$ of edge reweighting module, we design a clustering-based outlier detector to generate temporary predicted labels. Based on these pseudo labels, we raise a new outlier score function to guide the optimization process. The entire architecture operates in an end-to-end manner, allowing the model to learn both the optimal connections and the superior detected results simultaneously.

## 2.3 Clustering-based Outlier Detector

**Learnable Clustering Layer.** To preliminarily separate the normal cluster from the anomalous candidate cluster, we introduce a clustering layer (Guo et al., 2017) that incorporates learnable cluster centroids into our model. Specifically, the similarity between the latent representations $\mathbf{Z} = \{\mathbf{z}_1, \cdots, \mathbf{z}_N\}$ and the cluster centroids $\boldsymbol{\mu}$ is measured using the Student's $t$-distribution, yielding the soft clustering assignment probabilities $q$ for each sample across all clusters:

$$q_{ij} = \frac{(1 + \|\mathbf{z}_i - \boldsymbol{\mu}_j\|^2)^{-1}}{\sum_{j'=1}^c (1 + \|\mathbf{z}_i - \boldsymbol{\mu}_{j'}\|^2)^{-1}}, \tag{9}$$

where we assume that the $N$ samples are partitioned into $c$ classes. $c$ is set to 2, as our goal is to achieve binary separation between normal and anomalous nodes. The soft assignment probabilities $q_{ij}$ form a distribution matrix $Q$. To further refine the clustering process and improve the compactness of cluster assignments, the target distribution $P$ is defined as follows:

$$p_{ij} = \frac{q_{ij}^2 / \sum_{i=1}^N q_{ij}}{\sum_{j'=1}^c q_{ij'}^2 / \sum_{i=1}^N q_{ij'}}. \tag{10}$$

Given the target distribution $P$ and current distribution $Q$, we formulate the clustering loss as follows:

$$\ell_{\text{c}} = \text{KL}(P\|Q) = \sum_{i=1}^N \sum_{j=1}^c p_{ij} \log \frac{p_{ij}}{q_{ij}}. \tag{11}$$

This loss function serves to guide the clustering optimization process and encourages the learned embeddings to capture as much discriminative information as possible.

During the training stage, the predicted labels are computed via $\hat{y}_i = \arg\max_j(q_{ij})$ for the $i$-th instance. Then we first designate the cluster containing a relatively larger number of samples as the **normal cluster**, and **temporarily** treat all nodes within it as normal candidates. Conversely, the remaining cluster is considered the **anomalous candidate cluster**. This assumption is made based on the fact that, in most datasets for outlier detection tasks, normal data constitutes the majority. Generally, the learnable clustering module avoids the use of a pre-defined threshold by generating pseudo labels directly for outlier identification, thereby improving the reliability and robustness of the detection process.

**Diversity Loss.** However, the self-discriminative masking phase could encourage one cluster to contain no samples, resulting in a trivial solution, as this represents the easiest way to maximize the MMD value. We refer to this as class collapse of clustering, *i.e.*, the optimization process collapses

all nodes into a single cluster, thereby undermining the clustering effectiveness and impeding the self-discriminative masking phase. To address this, we design a regularization term:

$$\ell_{\text{diversity}} = \sum_{k=1}^{c} \max\left(0, \varepsilon - \hat{u}_k\right), \tag{12}$$

where $\hat{u}_k = \frac{1}{N} \sum_{i=1}^{N} q_{ik}$ represents the proportion of samples assigned to cluster $k$. $\varepsilon$ is a minimum proportion threshold that ensures each cluster maintains a sufficient number of samples. If $\hat{u}_k$ is greater than or equal to $\varepsilon$, the term becomes zero (*i.e.*, no penalty). Otherwise, it would become positive and penalize the whole objective loss.

**Inference.** Let $\mathcal{D}_{\text{pos}}$ and $\mathcal{D}_{\text{neg}}$ in Eq. (8) denote the sets of normal candidate nodes and anomalous candidate nodes, respectively. We define the anomalous scores according to the predicted logits in the clustering layer. Denoting $\boldsymbol{\mu}_{\text{pos}}$ as the centroid of the normal cluster, the anomaly score for node $i$ is defined as:

$$s_i = 1 - q_{i,\text{pos}} = 1 - \frac{\left(1 + \|\mathbf{z}_i - \boldsymbol{\mu}_{\text{pos}}\|^2\right)^{-1}}{\sum_{j'=1}^{c}\left(1 + \|\mathbf{z}_i - \boldsymbol{\mu}_{j'}\|^2\right)^{-1}}, \tag{13}$$

which means that the higher the scores of the nodes, the more anomalous they are. Combining all the above loss terms, we define the overall objective function as

$$L = \ell_{\text{r}} + \alpha \cdot \ell_{\text{c}} + \beta \cdot \ell_{\text{dr}} + \gamma \cdot \ell_{\text{diversity}}. \tag{14}$$

The proposed objective function enables the self-discriminative masking spoiler and outlier detector to be jointly optimized, facilitating the learning of a more discriminative latent representation. The reconstruction loss is employed to retain essential information from the original data. Concurrently, the self-discriminative loss encourages the nodes from the same cluster to strongly connect, thereby reducing anomalous information contamination. The detailed procedure of the proposed method and complexity analysis are summarized in Appendices B and C, respectively.

## 3 CONNECTION WITH PREVIOUS WORK

The concept of the "Homophily Trap" (He et al., 2024) crystallizes a long-standing challenge in graph anomaly detection. However, it measures anomaly degree using a pre-computed spectral property-based metric and then generates multi-level graph nodes, edges, and subgraphs accordingly, making the results heavily dependent on the quality of this metric. Previous attempts to mitigate it have primarily focused on graph rewriting (Dou et al., 2020; Liu et al., 2021a; Qiao & Pang, 2023; Gasteiger et al., 2019; Topping et al., 2022). These approaches, however, are often heavy-handed: they either risk destroying the graph's essential structure through complete reconstruction or rely on heuristics that require manually defined thresholds, raising concerns about reliability. A detailed introduction to these works can be found in Appendix I.

Our self-discriminative masking spoiler offers a more effective solution. Rather than rewriting connections, it adaptively **re-weights** them, preserving the original graph structure while surgically suppressing the information flow that causes the homophily trap. This targeted re-weighting differs fundamentally from the attention mechanism in GAT (Veličković et al., 2018). While GAT weights edges based on local feature similarity for representation learning, our masking spoiler is explicitly guided by a global, task-specific objective: maximizing the separation between clusters of normal and anomalous nodes. This guidance is a core component of our self-discriminative paradigm, which employs a clustering detector to generate pseudo-labels, creating a principled, end-to-end solution that is both adaptive and threshold-free.

## 4 EXPERIMENT

In this section, we provide a detailed the experimental settings, and conduct comprehensive experiments to answer the following research questions (RQs):

- **RQ1:** Does the proposed model outperform state-of-the-art graph outlier detection baselines?

Table 1: Average AUCs with standard deviation (10 trials) of different graph anomaly detection algorithms. The best and second-best results are **bolded** and underlined, respectively.

| Methods/Datasets | Email | Cora | Disney | Flickr | CiteSeer | Enron | Reddit | Amazon |
|---|---|---|---|---|---|---|---|---|
| L1SUB (Miller et al., 2010) | 72.89±0.26 | 52.53±0.00 | 59.80±4.73 | 54.43±0.02 | 63.86±0.13 | 60.85±2.51 | 56.94±0.00 | 47.53±0.02 |
| DEEPFD (Wang et al., 2018) | 52.84±0.00 | 51.96±0.00 | 50.28±0.40 | 52.94±0.00 | 52.25±0.00 | 50.00±0.00 | 51.67±0.00 | 50.00±0.00 |
| GAT+ClusterAD (Veličković et al., 2018) | 66.19±6.08 | 61.08±5.02 | 67.54±3.47 | 51.19±0.80 | 53.26±2.50 | 63.51±3.61 | 54.78±3.18 | 65.43±9.60 |
| DOMINANT (Ding et al., 2019) | 94.00±12.00 | 92.00±11.66 | 48.78±1.61 | 45.42±0.16 | 56.39±8.48 | 52.21±0.71 | 55.88±0.43 | 50.36±0.59 |
| AnomalyDAE (Fan et al., 2020) | 65.91±5.41 | 72.53±5.72 | 56.92±9.21 | 26.54±0.00 | 28.72±0.01 | 48.05±7.51 | 48.38±2.97 | 39.80±6.70 |
| CONAD (Xu et al., 2022) | 83.62±25.52 | 75.00±25.50 | 58.99±3.94 | 45.62±0.19 | 59.30±10.19 | 51.87±0.64 | 56.12±0.03 | 48.94±1.56 |
| AS-GAE (Zhang & Zhao, 2022) | 84.68±18.50 | 75.39±21.79 | 34.32±0.00 | 55.98±1.14 | 42.89±0.95 | 61.73±4.71 | 49.35±5.01 | 48.96±0.52 |
| TAM (Qiao & Pang, 2023) | 30.45±0.01 | 55.55±0.37 | 30.51±0.00 | 65.19±0.86 | 46.75±1.40 | 44.75±0.03 | 58.60±0.03 | 79.87±0.16 |
| ADA-GAD (He et al., 2024) | 81.85±10.76 | 71.68±0.01 | 41.10±5.25 | 55.99±0.02 | 68.08±0.00 | 59.94±4.69 | 56.17±0.10 | 50.86±1.10 |
| BOURNE (Liu et al., 2024) | 64.39±2.69 | 56.32±0.16 | 61.98±2.11 | 45.10±8.59 | 66.26±2.81 | 68.97±15.11 | 57.48±2.28 | 75.01±7.15 |
| GADAM (Chen et al., 2024) | 68.12±2.39 | **92.62±0.35** | 69.35±0.22 | 61.46±0.22 | **93.91±0.13** | 33.91±0.47 | 58.44±0.26 | 57.15±1.50 |
| AD-GCL (Xu et al., 2025) | 57.79±1.77 | 68.54±0.09 | 38.96±6.04 | 46.62±2.75 | 72.36±2.91 | 65.87±2.02 | 53.94±0.79 | 24.94±3.32 |
| SmoothGNN (Dong et al., 2025) | 51.09±15.37 | 62.72±8.14 | 54.40±8.64 | 50.26±4.26 | 52.82±1.99 | 52.35±3.40 | 58.38±6.23 | 49.96±0.08 |
| CER-GOD | **96.98±0.08** | 92.09±1.26 | **72.13±3.01** | **67.08±0.16** | 74.01±0.37 | **72.63±3.65** | **59.71±1.89** | **86.24±3.56** |

- **RQ2:** How do the hyper-parameters of the proposed method affect its detection performance?

- **RQ3:** Does the proposed method learn more effective and discriminative latent representations compared to other state-of-the-art methods?

- **RQ4:** Does the learned mask hold meaningful relevance?

- **RQ5:** What is the individual contribution of each component in the proposed method to graph anomaly detection?

## 4.1 EXPERIMENTAL SETTINGS

**Datasets.** We adopt eight datasets across five different types: citation networks, social networks, communication networks, organic and co-review, including Email, Cora, Flickr, CiteSeer, Disney, Enron, Reddit and Amazon. Detailed dataset descriptions are deferred to Appendix G.

**Implementation Details.** The implementation details are provided in Appendix H due to page limitations. To evaluate the anomaly detection performance of each method, we utilize the widely used metric: Area Under the Curve (AUC). The experimental results are reported as the mean and standard deviation, calculated over 10 independent runs of each algorithm to ensure a fair evaluation.

**Compared Baselines.** To evaluate the effectiveness of the proposed method, we compare it with two types of graph outlier detection baselines: ten node-level outlier detection methods, namely GAT+ClusterAD (Veličković et al., 2018), DOMINANT (Ding et al., 2019), AnomalyDAE (Fan et al., 2020), CONAD (Xu et al., 2022), ADA-GAD (He et al., 2024), BOURNE (Liu et al., 2024), AD-GCL (Xu et al., 2025), TAM (Qiao & Pang, 2023), SmoothGNN (Dong et al., 2025), and GADAM (Chen et al., 2024); and three subgraph-level outlier detection SOTAs: L1SUB (Miller et al., 2010), DEEPFD (Wang et al., 2018), and AS-GAE (Zhang & Zhao, 2022).

## 4.2 COMPARISON WITH STATE-OF-THE-ART BASELINES (RQ1)

Table 1 provides a detailed evaluation of the proposed method against thirteen recent baselines across eight widely used graph datasets. CER-GOD consistently achieves competitive performance across all datasets, attaining the best or second-best results in the majority of cases. For instance, on the Email dataset, our model surpasses the previous best (*i.e.*, AS-GAE) by more than 12%, indicating exceptional capability in identifying anomalies in communication networks. Moreover, the proposed method maintains consistently high accuracy and stable performance without catastrophic failures, reflecting both its robustness and adaptability to diverse graph characteristics. It is also worth noting that GAT+ClusterAD serves as a strong competitor, as it is constructed by integrating GAT with a clustering-based outlier detector. Nevertheless, CER-GOD achieves substantial improvements over it, further highlighting the effectiveness of the SD-MS module.

## 4.3 PARAMETER SENSITIVITY ANALYSIS (RQ2)

**Impact of Key Hyper-parameters $\alpha$, $\beta$ and $\gamma$.** To assess the influence of key hyper-parameters on the anomaly detection performance of the proposed model, we conduct a sensitivity analysis on the

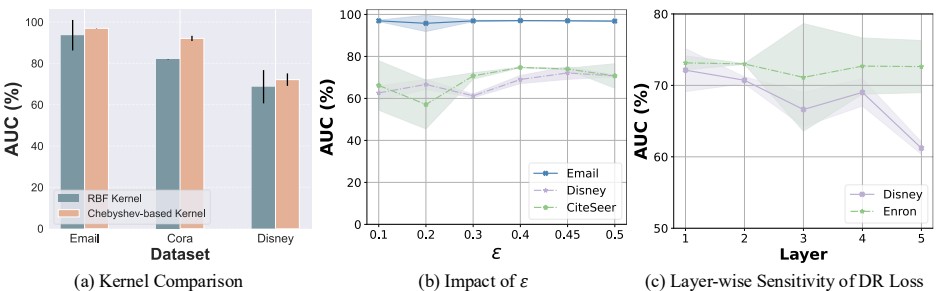

Figure 3: Parameter sensitivity of clustering loss coefficient $\alpha$, distribution repulsion loss coefficient $\beta$, and diversity loss $\gamma$.

Figure 4: Parameter sensitivities of 1) different kernel methods for MMD calculation; 2) impact of hyper-parameter $\epsilon$ that controls the minimum proportion of samples per cluster; 3) impact of applying distribution repulsion (DR) loss to different GCN layers.

hyper-parameters $\alpha$, $\beta$, and $\gamma$, which control the trade-off among the reconstruction loss, clustering loss, distribution repulsion loss, and penalty term. Figure 3 presents the AUC trends on the Flickr and Cora datasets across varying values $[1e-3, 1e3]$. The observations are shown as follows: (1) Too large hyper-parameters may cause the clustering loss, self-discriminative loss or penalty loss to dominate the reconstruction loss, which can lead to the learned embeddings losing their original semantic information, thereby increasing the difficulty of training; (2) Moderate values are beneficial for preserving semantic information while achieving better performance, as the multiple objectives compete with each other during training. (3) When the trade-off values are too small, the model tends to prioritize low-dimensional reconstruction at the expense of discriminative features, resulting in poor separation between normal and anomalous nodes and an underdeveloped decision boundary. (4) Our model exhibits a broad safe operating region (*e.g.*, $\alpha, \beta \in [0.01, 1]$) with consistently stable performance, enabling a reliable fixed default configuration.

**Impact of Different Kernel Methods for the MMD Calculation.** The results in Fig. 4(a) show that the Chebyshev-based kernel consistently outperforms the conventional RBF kernel across all three datasets. We attribute this improvement to the geometric properties of the metrics: the RBF kernel aggregates differences across all dimensions, which potentially dilutes anomalous deviations through a "smoothing" effect in high-dimensional spaces, while the Chebyshev kernel focuses exclusively on the maximum discrepancy along any single dimension. Given that anomalies generally exhibit as deviations in specific feature subsets rather than uniform global shifts, the Chebyshev kernel property enables the model to capture critical outlier patterns more effectively.

**Impact of Trade-off Parameter $\epsilon$.** We have included a detailed sensitivity analysis of the diversity loss hyperparameter $\epsilon$ in Fig. 4(b). It can be observed that excessively small values may be overly permissive and further enable cluster collapse, thereby compromising performance across all datasets. Instead, $\epsilon = 0.5$ denotes the strictest setting, which may slightly hurt performance because forcing an exact 50:50 balance may be too rigid for the nature of the data.

**Layer-wise Sensitivity Analysis for Distribution Repulsion Loss.** We conduct a sensitivity analysis for the distribution repulsion (DR) Loss by imposing it on different GCN layers (from 1st to

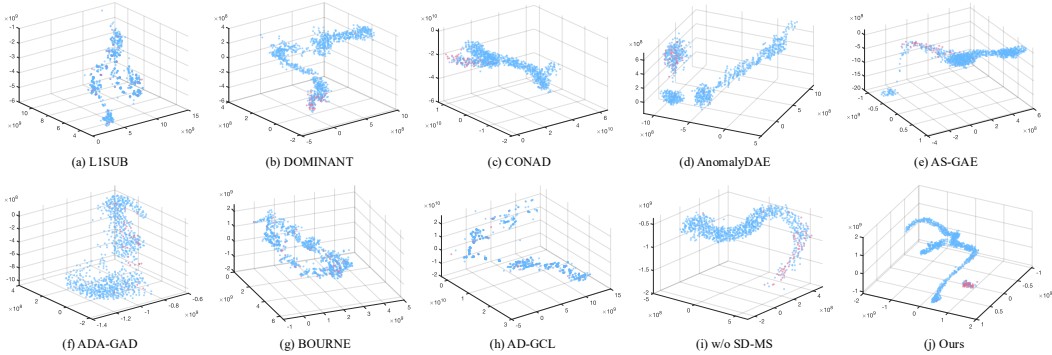

Figure 5: The comparison of t-SNE visualizations on the Email dataset for all baseline methods and the proposed model. Normal nodes are depicted in blue, while anomalous nodes are shown in red.

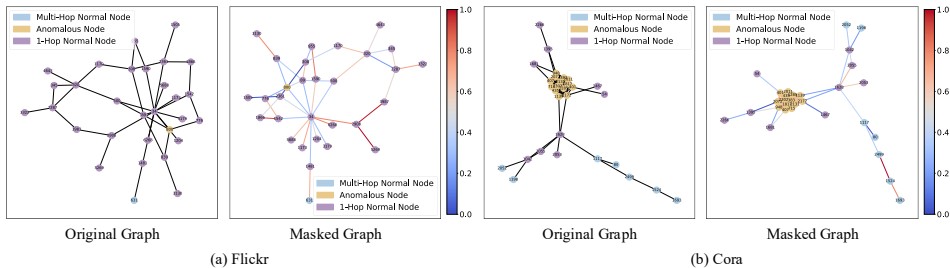

Figure 6: Visualization of the sampled subgraph topology on the Flickr and Cora datasets. The color bar represents the masking strength of edge connections.

5th) and compare the anomaly detection performance. Fig. 4(c) shows that performance consistently decreases as the layer depth increases from the 1st layer to the 5th layer, with the first layer yielding optimal results.

## 4.4 QUALITATIVE STUDY (RQ3&RQ4)

**Embedding Visualizations.** First, a more intuitive demonstration of the discriminative power of the learned embeddings is provided through t-SNE visualizations across eight baselines and the ablated model **w/o** Self-Discriminative Masking Spoiler (SD-MS), as shown in Figure 5. Compared to the baselines, the proposed model achieves a significantly better separation between normal and anomalous nodes. The anomalies are tightly clustered and clearly isolated from the majority of normal nodes, indicating that our model learns a more discriminative and structured embedding space. In contrast, most baseline methods, such as L1SUB, DOMINANT, and AD-GCL, show scattered or overlapping distributions of anomalous nodes, making them harder to detect. Even models like AS-GAE and CONAD, which display relatively better anomaly grouping, still fail to achieve the level of compactness and separation seen in our approach. The comparison with the degraded model (i) further confirms the effectiveness of the self-discriminative mask spoiler, as removing the SD-MS component results in more dispersed and less distinguishable anomaly embeddings.

**Mask Visualizations.** To intuitively illustrate the effectiveness of the learned mask, we display the masked graph structure (in Figure 6) and the learned mask (please refer to Appendix F). The visualized subgraph consists of 30 randomly sampled nodes, colored according to their corresponding classes. The key observations are as follows: **1)** The edge weights between inter-class nodes are significantly reduced, as indicated by edges predominantly colored in shades of blue. **2)** Edges connecting intra-class nodes tend to appear red. For example, in Subfigure (b), all anomalous nodes are interconnected with red edges. Although some normal nodes are linked by blue edges, they can still aggregate information through other undrawn normal nodes.

**Ablation Visualizations of SD-MS.** We also provide Figure 7 to present histograms of the $L_2$-norm distances between the learned node embeddings and vectors sampled from a standard Gaussian distribution $\mathcal{N}(\mathbf{0}, \mathbf{I}_k)$, comparing the cases with and without the proposed self-discriminative masking spoiler (SD-MS) module on Cora dataset (results on other datasets are presented in Appendix F due to page limitation). Here, the purple bars represent the distance distribution of normal nodes, while the blue bars correspond to anomalous nodes. This metric serves as a proxy to evaluate how well the embeddings of anomalous and normal nodes are separated in the latent space. The application of the SD-MS module significantly increases the separation between the distance distributions of normal and anomalous nodes. Specifically, in all datasets, SD-MS reduces the overlap between the two classes, resulting in more distinguishable and polarized score distributions: normal nodes cluster near lower distances, while anomalous nodes shift toward higher distances.

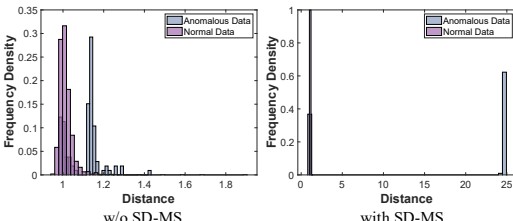

Figure 7: Distribution histograms of embedding distances with or w/o SD-MS on Cora. The distance is computed between learned embeddings and vectors sampled from a standard Gaussian distribution $\mathcal{N}(\mathbf{0}, \mathbf{I}_k)$ through $L_2$-norm.

## 4.5 ABLATION STUDY (RQ5)

Here, we conduct an ablation study to demonstrate the effectiveness of each component in the proposed framework. Specifically, we design three degraded modules, including **1) w/o** Reconstruction variant (which applies the learnable clustering directly to the GCN encoder), **2)** Reconstruction-based Outlier Detector (*i.e.*, **w/o** Clustering-based Outlier Detector); **3)**

Table 2: Ablation Study on Email, Cora, and Flickr (mean (%)±std (%)).

| Methods/Datasets | Email | Cora | Flickr |
|---|---|---|---|
| **w/o** Reconstruction | 50.80±3.10 | 52.20±14.33 | 49.24±2.35 |
| Reconstruction OD | 84.12±1.11 | 79.51±1.60 | 56.70±5.15 |
| **w/o** SD-MS | 87.05±4.66 | 78.84±2.68 | 58.40±6.61 |
| Ours | **96.98±0.08** | **92.09±1.26** | **67.08±0.16** |

**w/o** Self-Discriminative Masking Spoiler (SD-MS). Note that we utilize the percentile filter strategy to get the pseudo labels for the degraded model Reconstruction OD, which is set to the actual anomaly ratio of all samples. The experimental results are shown in Table 2, which clearly demonstrate that the full model consistently outperforms all degraded variants across datasets, confirming the contributions of the clustering-based OD as well as the SD-MS module. In contrast, replacing the detection mechanism with reconstruction error yields significantly inferior performance, underscoring the necessity of the proposed components for effective anomaly detection. The results also show that removing the reconstruction component leads to a significant performance drop, which suggests the necessity of this component in our method. Besides, please note that the diversity loss cannot be ablated, as it is essential for preventing class collapse in the clustering layer.

## 5 CONCLUSION

In this work, we addressed the "Homophily Trap", a fundamental challenge in graph outlier detection where message-passing mechanisms of graph convolution operation inadvertently blur the distinction between normal and anomalous nodes. Our proposed end-to-end framework successfully dismantles this trap by synergistically combining several key innovations. The core of our approach is a self-discriminative masking spoiler that intelligently re-weights graph connections, effectively filtering out contaminating information from dissimilar neighbors without destroying the underlying graph structure. This masking process is guided in a fully unsupervised manner by an adaptive clustering-based detector, which provides crucial pseudo-labels and frees the model from relying on arbitrary thresholds. To maintain stability and prevent the common pitfall of class collapse during optimization, we integrated a diversity loss. The joint optimization of these elements allows our model to learn a powerfully discriminative latent space, culminating in state-of-the-art performance on a wide range of benchmark datasets. The limitation is that this work currently focuses on binary anomaly detection; multi-class extension remains a direction for future work.

ACKNOWLEDGMENTS

The work is funded by the Science and Technology Development Fund (FDCT), Macau SAR (File No. 001/2024/SKL, 0072/2025/AMJ, 0123/2023/RIA2, CG2026-IOTSC). This research is also supported by the National University of Singapore, Institute of Data Science. Any opinions, findings and conclusions or recommendations expressed in this material are those of the author(s) and do not reflect the views of the National University of Singapore.

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

## A  PROOF OF PROPOSITION 1

*Proof.* Assume that the shortest path between node $i$ and $j$ is $r$, the gradient of activation function in the $l$-th layer follows $|\nabla \sigma_l| \leq \alpha$, the gradient of aggregating function at layer $l$ satisfies $|\nabla \text{Aggr}_l| \leq \beta$ and $\mathbf{z}_i^{(0)} = \mathbf{x}_i$, then the Jacobian at layer $l$ is bounded by:

$$\left| \frac{\partial \mathbf{z}_j^{(l)}}{\partial \mathbf{x}_i} \right| \leq (\alpha\beta)^l (\mathbf{A}^l)_{ji}. \tag{15}$$

Next step, let $\mathcal{N}(j)$ be the neighbor set of node $j$, we have the Jacobian at layer $l+1$ via the chain rule for differentiation:

$$\frac{\partial \mathbf{z}_j^{(l+1)}}{\partial \mathbf{x}_i} = \frac{\partial \sigma_l \left( \text{Aggr}_l \left( \left\{ \mathbf{z}_k^{(l)} : k \in \mathcal{N}(j) \cup \{j\} \right\} \right) \right)}{\partial \text{Aggr}_l} \cdot \frac{\partial \text{Aggr}_l \left( \mathbf{z}_{\mathcal{N}(j)}^{(l)} \right)}{\partial \mathbf{z}_k^{(l)}} \cdot \frac{\partial \mathbf{z}_k^{(l)}}{\partial \mathbf{x}_i}. \tag{16}$$

Thus, we will obtain:

$$\left| \frac{\partial \mathbf{z}_j^{(l+1)}}{\partial \mathbf{x}_i} \right| \leq \alpha\beta \sum_{k \in \mathcal{N}(j)} \left| \frac{\partial \mathbf{z}_k^{(l)}}{\partial \mathbf{x}_i} \right|, \tag{17}$$

and the bound would become:

$$\left| \frac{\partial \mathbf{z}_j^{(l+1)}}{\partial \mathbf{x}_i} \right| \leq \alpha\beta \sum_{k \in \mathcal{N}(j)} (\alpha\beta)^l (\mathbf{A}^l)_{ki}. \tag{18}$$

Herein, $\mathbf{A}^l$ represents the adjacency matrix raised to the layer $l$, the sum $\sum_{k \in \mathcal{N}(j)} (\mathbf{A}^l)_{ki}$ essentially counts the number of paths of length $l$ from node $i$ to node $j$. This can be further expressed as:

$$\left| \frac{\partial \mathbf{z}_j^{(l+1)}}{\partial \mathbf{x}_i} \right| \leq (\alpha\beta)^{l+1} (\mathbf{A}^{l+1})_{ji}, \tag{19}$$

where $(\mathbf{A}^{l+1})_{ji} = \sum_k \mathbf{A}_{jk} (\mathbf{A}^l)_{ki}$ and $\mathbf{A}_{jk} = 1$ due to $k \in \mathcal{N}(j)$. As $r$ is the shortest distance between node $i$ and $j$, then the inequality becomes:

$$\left| \frac{\partial \mathbf{z}_j^{(r+1)}}{\partial \mathbf{x}_i} \right| \leq (\alpha\beta)^{r+1} (\mathbf{A}^{r+1})_{ji}. \tag{20}$$

Then finished the proof. □

## B  DETAILED ALGORITHM PROCEDURES

Here, we present the detailed procedures for CER-GOD in Algorithms 1.

## C  COMPLEXITY ANALYSIS

Assume that there is a graph of $N$ nodes, each with feature dimension $d$, $E$ edges, and latent dimension $k$, the complexity of the overall framework is computed as follows:

- The whole encoder adopts an $L$-layer GCN backbone, where each layer involves two key operations: neighbor aggregation ($\mathcal{O}(E)$) and linear transformation (each layer has a complexity of $\mathcal{O}(N \cdot d \cdot k)$).
- The decoder typically involves a linear transformation to reconstruct the input feature matrix, where complexity is $\mathcal{O}(N \cdot d \cdot k)$. The complexity of computing topology $\hat{\mathbf{A}}$ is $\mathcal{O}(N^2 \cdot k)$.
- The clustering layer computes the similarity between each node's embedding and the cluster centroids and further calculates the KL Divergence loss. Hence, the complexity is $\mathcal{O}(N \cdot c)$, where $c$ is the class number.

---

**Algorithm 1** Clustering-guided Edge Reweighting for Graph Outlier Detection (CER-GOD)

---

**Input:** The input graph set $\mathcal{G}$, dimensions of GCN hidden layers $k$, trade-off parameters of clustering loss $\alpha$ and distribution repulsion loss $\beta$, and diversity loss $\gamma$, learning rate $\eta$.

**Output:** The outlier detection scores $\mathbf{s}$.

1: Initialize the parameters $\mathbf{M}$ and $\Phi$ of self-discriminative masking spoiler and clustering-based outlier detector network parameters $f_\Phi$;
2: Reallocate topology through Eq. (5) and normalized it with $\tilde{\mathbf{A}} + \mathbf{I}_N$;
3: Initialize the cluster centroids $\boldsymbol{\mu}$ via performing $K$-Means on latent representation $\mathbf{Z} = f_\mathcal{W}^{\mathrm{enc}}(\mathbf{X}, \tilde{\mathbf{A}})$;
4: **while** not convergence **do**
5:     Obtain the latent $\mathbf{Z}$ and $\mathbf{Z}^{(0)}$ by $\mathbf{Z} = f_\mathcal{W}^{\mathrm{enc}}(\mathbf{X}, \tilde{\mathbf{A}})$ and Eq. (2);
6:     Calculate the reconstruction loss via Eq. (4);
7:     Compute the current cluster assignment distribution $Q$ by Eq. (9);
8:     Compute the target cluster assignment distribution $P$ by Eq. (10);
9:     Calculate the diversity loss via Eq. (12);
10:    Compute the cluster label for sample $i$ via $\hat{y}_i = \arg\max_j(q_{ij}), i = 1, 2, \ldots, N$;
11:    Calculate the distribution repulsion loss based on Eq. (8);
12:    Calculate total loss via Eq. (14);
13:    Back-propagate masking spoiler and outlier detector networks and update $\mathbf{M}$, $\Phi$, and $\boldsymbol{\mu}$ respectively;
14: **end while**
15: Calculate the outlier detection scores $\mathbf{s}$ through Eq. (13);
16: **return** The outlier detection scores $\mathbf{s}$.

---

- The diversity loss is computed by calculating the proportion of samples assigned to each cluster, then applying a penalty term. The complexity of this operation is $\mathcal{O}(N \cdot c)$.

- For $\ell_{\mathrm{dr}}$, assume that positive set $\mathcal{D}_{\mathrm{pos}}$ contains $m$ nodes, intra-calculation consumes $\mathcal{O}(m^2 \cdot k)$ and $\mathcal{O}((N - m)^2 \cdot k)$. Inter-calculation consumes $\mathcal{O}(m \cdot (N - m) \cdot k)$. In the worst case, it would be $\mathcal{O}(N^2 \cdot k)$.

Thus, the total computational complexity is $\mathcal{O}(L \cdot (E + N \cdot k \cdot d) + N \cdot c + N^2 \cdot k)$.

We also provide the running time comparison in Table 3 for reference. Across all four datasets, CER-GOD demonstrates highly competitive efficiency, consistently ranking among the fastest methods. On Email, Cora, and Disney, CER-GOD achieves second-tier performance, running notably faster than most baselines. These results confirm that CER-GOD provides an effective balance between accuracy and efficiency, making it a practical choice for large-scale anomaly detection.

Table 3: Running time (in seconds) comparison on four datasets, the results are recorded at the time of running 200 epochs for fairness.

| Methods/Datasets | Email | Cora | Disney | CiteSeer |
|---|---|---|---|---|
| DOMINANT (Ding et al., 2019) | 5.6278 | 17.7207 | 1.4391 | 49.0877 |
| AnomalyDAE (Fan et al., 2020) | 4.0151 | 6.1372 | 0.9438 | 24.4646 |
| CONAD (Xu et al., 2022) | 9.0529 | 33.0815 | 2.9236 | 73.6654 |
| AS-GAE (Zhang & Zhao, 2022) | 15.9610 | 24.7159 | 2.8146 | 27.9659 |
| ADA-GAD (He et al., 2024) | 52.4239 | 321.3380 | 7.2619 | 1258.2931 |
| BOURNE (Liu et al., 2024) | 11.5739 | 21.2274 | 5.9707 | 30.1851 |
| AD-GCL (Xu et al., 2025) | 1991.0725 | 5510.4095 | 116.0679 | 3223.6459 |
| CER-GOD | 42.4534 | 82.8717 | 6.7685 | 101.0798 |

## D EXPERIMENT ON LARGE-SCALE GRAPH BENCHMARK

We evaluated CER-GOD on the OGB-Proteins dataset, which contains **132,534** nodes and **79,164,284** edges, where CER-GOD is compared with 5 state-of-the-art baselines: DOMINANT (Ding et al., 2019), ComGA (Luo et al., 2022), SL-GAD (Zheng et al., 2021), CoLA (Liu et al., 2021b) and TAM (Qiao & Pang, 2023). As shown in the Table 4 (note that we reported the performance of all baseline

methods directly from (Qiao & Pang, 2023)), CER-GOD scales successfully and outperforms several state-of-the-art baselines in comparison.

Table 4: AUCs (%) of different graph anomaly detection algorithms on large-scale dataset OGB-Proteins. The best result is **bolded**.

| Methods/Datasets | DOMINANT | ComGA | CoLA | SL-GAD | TAM | CER-GOD |
|---|---|---|---|---|---|---|
| OGB-Proteins | 72.67 | 71.34 | 71.42 | 73.71 | 74.49 | **74.81** |

## E    DISCUSSION OF CLUSTER NUMBER SELECTION

In this paper, we focus on the unsupervised outlier detection task. In a fully unsupervised setting, we lack labels to map multiple learned clusters to the binary ground truth. For example, if we set $c = 5$, determining which specific subsets of clusters represent "normal" and which represent "anomalous" requires additional heuristics (*e.g.*, "the smallest 3 clusters are anomalies"), which introduces significant instability and potential bias. Therefore, we followed the standard one-class classification setting and set the number of clusters as $c = 2$, as many previous works (Ding et al., 2019; Fan et al., 2020; Xu et al., 2022; He et al., 2024; Xu et al., 2025) did: The "anomalous candidate cluster" does not assume all anomalies are homogeneous, but instead treats anomalous patterns (*e.g.*, point/structural/attribute anomalies) as different manifestations of deviating from the manifold with respect to the normal pattern.

Regarding computational cost, as analyzed in Appendix C, the reconstruction process is indeed a major time-consuming component in our method. Compared to the reconstruction component, the clustering overhead is linear to $N$ and proportional to the small $c$ (here $c = 2$). Thus, adding clustering does not change the overall asymptotic time complexity and adds only a modest overhead in practice.

## F    SUPPLEMENTARY VISUALIZATION RESULTS

**Homophily Trap Visualizations.**    Here, we supplement four additional histograms of maximum mean discrepancy distances on two synthetic datasets (Attribute Anomaly and Structure Anomaly), and two real-world datasets (Cora and Flickr) in Fig. 8. The distance is computed between the standard Gaussian distribution $\mathcal{N}(\mathbf{0}, \mathbf{I}_d)$ and three types of node embeddings (normal node multi-hop away from anomalies, normal node 1-hop away from anomalies, and anomalous nodes).The embeddings are obtained through a single-layer graph convolution operation without any additional linear projection layer, isolating the pure effects of graph convolution. The synthetic datasets, derived from (Zhang & Zhao, 2022), demonstrate the homophily trap phenomenon in data containing either attribute anomalies or structure anomalies alone. Real-world data further exhibits both anomaly types simultaneously. This confirms the presence of the 'homophily trap' in real-world applications: graph convolution significantly influences the embeddings of neighboring normal nodes, causing them to become nearly indistinguishable from 1-hop anomalous nodes.

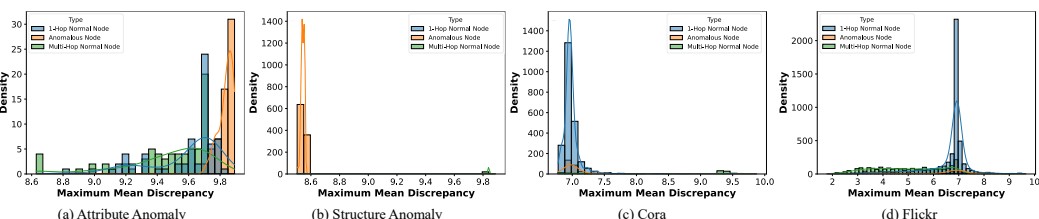

(a) Attribute Anomaly       (b) Structure Anomaly       (c) Cora       (d) Flickr

Figure 8: The histograms of maximum mean discrepancy distances on two synthetic datasets and two real-world datasets.

**Embedding Visualizations.**    Figure 9 presents 3D scatter plots comparing the performance of various anomaly detection methods on the Cora dataset. The blue points represent normal data, and

the pink points represent anomalies. Overall, the proposed method CER-GOD achieves the best separation between normal and anomalous points, demonstrating superior performance compared to other methods such as L1SUB, DOMINANT, and CONAD, which show less distinct separation. While methods like AS-GAE and ADA-GAD improve anomaly detection, the proposed approach clearly outperforms all, providing the most effective distinction between normal and anomalous data.

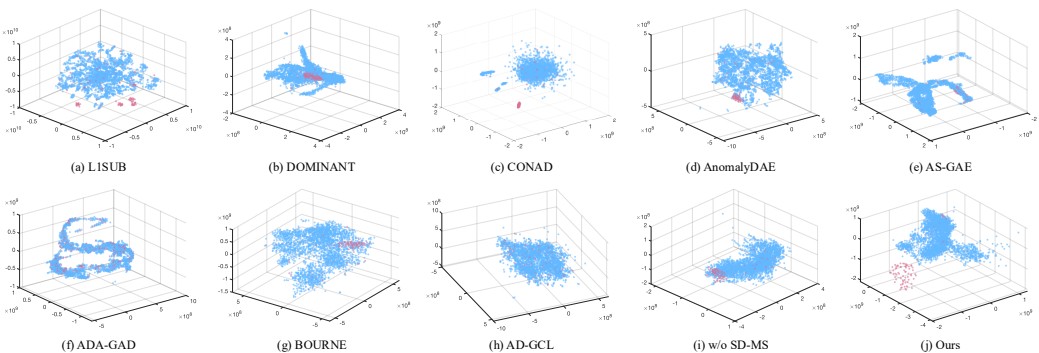

Figure 9: The comparison of t-SNE visualizations on the Cora dataset for all baseline methods and the proposed model. Normal nodes are depicted in blue, while anomalous nodes are shown in red.

**Impact of GNN Backbone.** We also conduct a parameter analysis of the adopted different GNN backbones in Figure 10, which includes GCN (Welling & Kipf, 2016), GIN (Xu et al., 2019), and GAT (Veličković et al., 2018). This figure compares the AUC performance of the above-mentioned backbones on three datasets: Cora, Email, and CiteSeer. GCN outperforms the others across Cora and Email, with the largest gap on Cora. GAT follows closely, especially on CiteSeer and Email, while GIN shows the weakest performance, particularly on Email. While GCN and GIN show similar performance with no significant gap, GAT tends to perform weaker. This is likely because GAT already incorporates masking in its mechanism, which

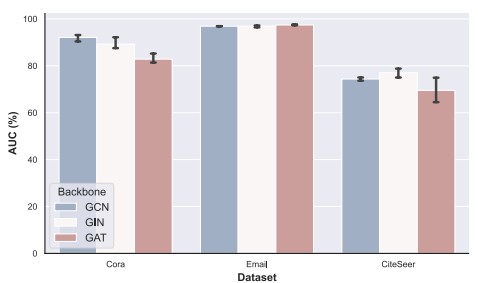

Figure 10: The performance comparison of different adopted backbones on the Cora, Email, and CiteSeer datasets.

adds an additional optimization burden to the self-discriminative module. It is worth noting that all three backbones outperform the other baselines.

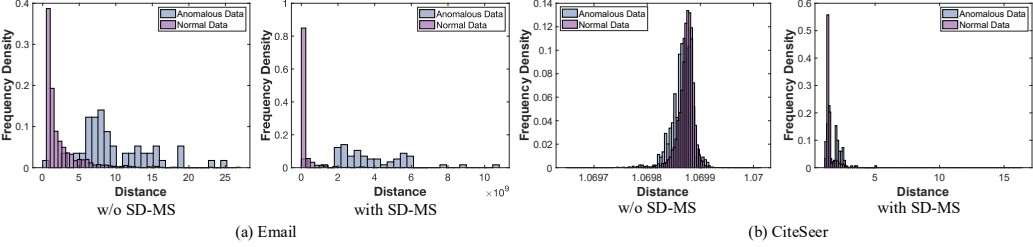

Figure 11: Distribution histograms of embedding distances with or w/o SD-MS on Email and CiteSeer datasets. The distance is computed between learned embeddings and vectors sampled from a standard Gaussian distribution $\mathcal{N}(\mathbf{0}, \mathbf{I}_{d'})$ through $L_2$-norm.

**Ablation Visualizations of SD-MS.** Similar conclusions can be observed in Figure 11. The area of the overlapping region obviously decreases after the masking procedure. Also, the distance distributions of normal and anomalous nodes are more discriminative intuitively.

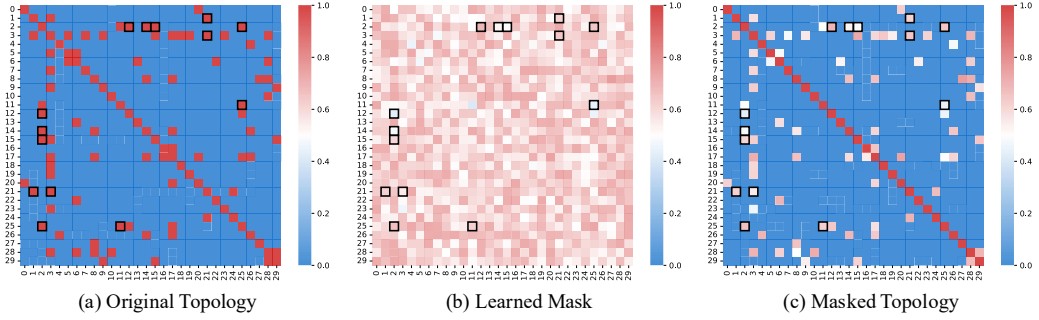

|          |          |          |
|:--------:|:--------:|:--------:|
| (a) Original Topology | (b) Learned Mask | (c) Masked Topology |

Figure 12: Visualization of the learned mask in the proposed model on the Flickr dataset. The color bar represents the masking strength of edge connections. Connections between different-class node pairs are highlighted with black frames.

**Mask Visualizations.** The learned mask and its impact are illustrated in Figure 12. A subgraph consisting of 30 randomly selected nodes is shown, with elements corresponding to different-class node pairs highlighted in a black frame. It is clearly evident that the relationships framed in the figure are significantly weakened, as most of them are close to white or even in blue, indicating high masking strength. For example, the relationships between Node 12 and Node 2, as well as between Node 25 and Node 11, exhibit such weakened connections.

## G  DATASETS

The detailed statistical descriptions are shown in Table 5. We also report the number of edges with "Homophily Trap" (HT) for the reference. Specifically, we adopt the Email and Cora in (Zhang & Zhao, 2022), CiteSeer, Flickr in (Pan et al., 2023), Amazon, OGB-Proteins in (Qiao & Pang, 2023), and Disney, Enron, Reddit in PyGOD[1].

Table 5: Detailed information of the graph benchmark datasets.

| Dataset | # Nodes | # Edges | # Average $[E]$ | # Edges w/o HT | # Average $[E]$ w/o HT | Anomaly Rate | Data Types |
|---|---|---|---|---|---|---|---|
| Email | 1,005 | 28,275 | 28.1300 | 22,190 | 22.0796 | 6.01% | Communication Network |
| Flickr | 7,575 | 482,555 | 63.7036 | 428,301 | 56.5413 | 6.24% | Social Network |
| Cora | 2,708 | 15,045 | 5.5557 | 14,383 | 5.3112 | 4.07% | Citation Network |
| CiteSeer | 3,327 | 10,275 | 3.0883 | 9,563 | 2.8744 | 4.72% | Citation Network |
| Disney | 124 | 335 | 2.7016 | 318 | 2.5645 | 4.80% | Organic |
| Enron | 13,533 | 176,987 | 13.0782 | 176,794 | 13.0639 | 0.04% | Organic |
| Reddit | 10,984 | 168,016 | 15.2964 | 159,688 | 14.5382 | 3.30% | Organic |
| Amazon | 10,244 | 351,216 | 34.3521 | 283,594 | 27.7381 | 6.66% | Co-review |
| OGB-Proteins | 132,534 | 79,164,284 | 597.3130 | 79,164,284 | 597.3130 | 4.50% | Biology Network |

## H  DETAILED EXPERIMENTAL SETTINGS

- **Trade-off Parameters:** For the proposed method, there are three critical hyper-parameters, $\alpha$, $\beta$, and $\gamma$, in their loss functions, which control the contributions of the clustering loss and distribution repulsion loss, and diversity loss, respectively. Section 4.3 includes the evaluation of the impact of variations in the values of all hyper-parameters on the anomaly detection performance. For our main experiments, the complete hyperparameter configurations are summarized in Table 6.

- **Baseline Settings:** All baselines are reproduced via publicly available code with their default parameter settings. Particularly, we employ the same architecture of the backbone network as the proposed method to ensure a fair comparison.

- **Training Details:** We utilize Adam (Kingma & Ba, 2014) for training. Besides, we set the learning rate $\eta$ to 5e-5 with the total training epochs to 300. For the proposed model, a 3-layer GCN backbone is adopted for the outlier detector, and the hidden dimension is set to 16.

---

[1]https://github.com/pygod-team/data

- **Implementation:** All experiments are executed on the NVIDIA Tesla H100 GPU (80 GB) with Intel Xeon Platinum 8480CL CPU.

Table 6: Detailed hyper-parameter settings on all datasets.

| Hyper-parameters | Email | Cora | Disney | Flickr | CiteSeer | Enron | Reddit | Amazon |
|---|---|---|---|---|---|---|---|---|
| $\alpha$ | 0.001 | 0.1 | 0.001 | 0.1 | 0.001 | 0.001 | 10 | 10 |
| $\beta$ | 0.001 | 10 | 0.01 | 0.001 | 1 | 1 | 0.001 | 0.1 |
| $\gamma$ | 1 | 1 | 1 | 1 | 1 | 1 | 1 | 1 |

## I  RELATED WORK

### I.1  GRAPH NEURAL NETWORK

Graph neural networks (GNNs) have emerged as a dominant framework for modeling graph-structured data, owing to their ability to aggregate information from a node's neighbors based on the underlying topology. Over the past decade, a wide range of architectural innovations have been introduced to improve their expressivity, efficiency, and generalization, spanning foundational models such as GCN, GraphSAGE, and GIN to more recent advances like InfoGraph and kernel-based graph learning (Welling & Kipf, 2016; Hamilton et al., 2017; Xu et al., 2019; Sun et al., 2019; 2023; Cai et al., 2024; 2026). GNNs are applicable to both node-level and graph-level tasks, with the latter often relying on readout operations to map a graph into a vector space. Recent research has further expanded the landscape (Scholkemper et al., 2025; Pahng & Hormoz, 2025; Sun et al., 2024). For example, Luo et al. (2025) introduced GNN+, showing that classical GNNs augmented with edge features, normalization, dropout, residual connections, feed-forward layers, and positional encodings can match or outperform graph Transformers on graph-level tasks while being more efficient; Papillon et al. (2025) also presented TopoTune, extending GNNs to topological deep learning via Generalized Combinatorial Complex Neural Networks that capture higher-order interactions with reduced complexity.

### I.2  GRAPH OUTLIER DETECTION

Node-level outlier detection on graphs (Qiao et al., 2025; Zhao et al., 2025; Pan et al., 2025) has developed through several methodological paradigms, each emphasizing different aspects of structural and attribute information. Reconstruction-based methods (Ding et al., 2019; Fan et al., 2020; Cai et al., 2025a;b) aim to rebuild node attributes or graph structures using autoencoders, flagging nodes with high reconstruction errors as anomalies. Prediction-based methods (*e.g.*, Li et al. (2017)) infer node features or links based on neighbors or learned patterns, where deviations from predictions indicate abnormality. Contrastive learning-based methods (Liu et al., 2021b; Dillon et al., 2024; Xu et al., 2025; Sun & Fan, 2024) distinguish normal and abnormal behaviors by learning robust node representations through contrasting positive and negative samples, thereby enlarging margins between inliers and outliers. Distance or deviation-based methods (Chen et al., 2020; Breunig et al., 2000; Zhang et al., 2024) measure statistical divergence in local neighborhoods, leveraging density or clustering cues to spot anomalies. Beyond these categories, hybrid and advanced designs have emerged: adversarial approaches like AS-GAE (Zhang & Zhao, 2022) enhance discrimination via perturbation; boundary-aware representations such as BOURNE (Liu et al., 2024) and adaptive augmentation in ADA-GAD (He et al., 2024) refine normality decision boundaries; and interpretable designs like CONAD (Xu et al., 2022) integrate attention for anomaly explanation. Recent efforts also target robustness under homophily and heterophily by incorporating denoising strategies, geometric embeddings (*e.g.*, hyperbolic spaces), and transformer-style encoders to mitigate over-smoothing and over-squashing. Collectively, these approaches represent a progression from early subspace and residual scoring to more sophisticated adversarial, contrastive, and one-class objectives, reflecting the field's ongoing pursuit of greater accuracy, interpretability in node-level anomaly detection.

### I.3  GRAPH REWRITING FOR GRAPH MINING

For the homophily trap issue, graph rewriting is a direct solution that aims at breaking the connection between inter-class nodes while keeping intra-class connections. Dou et al. (2020) raised a label-

aware similarity measure to identify informative neighbors, use reinforcement learning to determine the optimal number to select, and aggregate the chosen neighbors across different relations. Liu et al. (2021a) proposed that for the fraud target node, the redundant links could be filtered by choosing neighbors that are far from the target, measured by the distance, and removing them from the neighbor set. And the necessary links, which are beneficial for fraud prediction, would be created by choosing similar nodes of the fraud class and regarding them as neighbors. Qiao & Pang (2023) calculated the Euclidean distance and removed the relatively farther neighbor nodes for one node, finally adopting the similarity of nodes to detect anomalies. Gasteiger et al. (2019) designed the Graph Diffusion Convolution to aggregate information from a larger neighborhood by constructing it through a new graph, generated by sparsifying a generalized form of graph diffusion. Topping et al. (2022) introduced a new combinatorial edge-based curvature, the Balanced Forman curvature, which provides a sharp lower bound to the standard Ollivier curvature on graphs, and demonstrated that negatively curved edges contribute to this phenomenon. The limitations of current works have been discussed in Section 3.

## J    REPRODUCIBILITY STATEMENT

To ensure full reproducibility, our source code is provided in the supplementary material. We also plan to release the curated training dataset and final model weights. The experimental framework is described throughout the paper for transparency, and details on hardware details, model configurations, and hyper-parameters settings can be found in Appendix H. Dataset resources are summarized in Appendix G, and all evaluation benchmarks, which are publicly available, are listed in Section 4.1.

## K    THE USE OF LARGE LANGUAGE MODELS (LLMS)

The conceptual framework and core ideas outlined in this paper represent the authors' original contributions. AI-driven language models were employed solely as auxiliary tools to support specific well-defined tasks. These tasks encompassed implementing basic utility functions, and assisting with manuscript translation and linguistic polishing. The authors take full responsibility for the content of the manuscript, including any text generated or polished by the LLM. We have ensured that the LLM-generated text adheres to ethical guidelines and does not contribute to plagiarism or scientific misconduct.

