# OpenReview forum: "Escaping the Homophily Trap: A Threshold-free Graph Outlier Detection Framework via Clustering-guided Edge Reweighting"
_ICLR.cc/2026/Conference — ICLR 2026 Poster_

### Official Review · Reviewer_6mWp · 2025-10-18

**Soundness:** 2
**Presentation:** 2
**Contribution:** 2
**Rating:** 2
**Confidence:** 4

**Summary:**

The authors propose a Clustering-guided Edge Reweighting framework for Graph Outlier Detection (CER-GOD), which jointly optimizes a self-discriminative masking spoiler with an adaptive clustering-based outlier detector. Experiments show their improvement to some extent.

**Strengths:**

1. The authors propose a cluster-based edge reweighting framework for unsupervised graph outlier detection.
2. Experiments show their improvement to some extent.

**Weaknesses:**

1. Although the authors provide a time complexity, the N-squared complexity might be a hindrance for real deployment. It will be better if the authors can provide an experimental comparison.
2. The included datasets are relatively small compared to the newest anomaly (outlier) detection datasets, such as those in [1]. The authors should include the real-world datasets in their comparison.
3. The included baselines are not comprehensive, which questions the effectiveness of the framework. The authors should include SOTA works such as [2], [3], and [4].
4. Figure 3 shows that the method can be very sensitive to the variation of hyperparameters. As stated in Appendix G, they adopt grid search for finding the hyperparameters. It can be a question of how to utilize grid search for unsupervised learning without any ground truth labels. Furthermore, it can be difficult for the method to choose proper hyperparameters to operate on new datasets.
5. The authors should show the finetuned hyperparameters and how many times they conduct the experiments to search for the hyperparameters.

[1] Jianheng Tang, Fengrui Hua, Ziqi Gao, Peilin Zhao, Jia Li. GADBench: Revisiting and Benchmarking Supervised Graph Anomaly Detection. NeurIPS 2023.

[2] Hezhe Qiao, Guansong Pang. Truncated Affinity Maximization: One-class Homophily Modeling for Graph Anomaly Detection. NeurIPS 2023.

[3] Jingyan Chen, Guanghui Zhu, Chunfeng Yuan, Yihua Huang. Boosting Graph Anomaly Detection with Adaptive Message Passing. ICLR 2024.

[4] Xiangyu Dong, Xingyi Zhang, Yanni Sun, Lei Chen, Mingxuan Yuan, Sibo Wang. SmoothGNN: Smoothing-aware GNN for Unsupervised Node Anomaly Detection. WWW 2025.

**Questions:**

Please refer to the weaknesses.

---

> ### Author Response · Authors · 2025-11-21
>
> **We thank the reviewer for the comments. Please find our responses to your concerns below.**
>
> **Response to W1:** Follow your suggestion, we have supplemented the running time comparison is shown as follows:
>
> $\begin{matrix}
>     \hline
>  && \text{Email} & \text{Cora} & \text{Disney} & \text{CiteSeer} \\\\ \hline
> & \text{DOMINANT} & 5.6278& 17.7207& 1.4391&49.0877\\\\
> & \text{AnomalyDAE} & 4.0151 &6.1372 & 0.9438&24.4646\\\\
> & \text{CONAD} &9.0529 &33.0815 & 2.9236&73.6654\\\\
> \text{Running Time (in Seconds)}& \text{AS-GAE} & 15.9610&24.7159 & 2.8146 & 27.9659 \\\\
> & \text{ADA-GAD} &52.4239 &321.3380 &7.2619 &1258.2931 \\\\
> & \text{BOURNE} & 11.5739 &21.2274 &5.9707 & 30.1851 \\\\
> & \text{AD-GCL} & 1991.0725&5510.4095 &116.0679 &3223.6459\\\\
> & \text{CER-GOD} & 42.4534 &82.8717 & 6.7685&101.0798\\\\
> \hline
> \end{matrix}$
>
> It can be concluded that across all four datasets, CER-GOD demonstrates competitive efficiency. On Email, Cora, and Disney, CER-GOD achieves second-tier performance, running faster than most baselines. These results, together with the SOTA performance shown in **Table 1**, demonstrate that the CER-GOD achieves an effective balance between accuracy and efficiency.
>
> **Response to W2:** Thank you for your suggestion, and we would like to address your concern via the following perspectives:
>
> - Following your advice, we have supplemented a new experiment on a large-scale real-world dataset (OGB-Proteins), which contains **132,534 nodes** and **79,164,284 edges**. As shown in the table below, CER-GOD outperforms several SOTA baselines (note that we reported the performance of all baseline methods directly from[1]).
>
> $\begin{matrix}
>     \hline
>     &\text{DOMINANT} &\text{ComGA} & \text{CoLA}& \text{SL-GAD} & \text{TAM} & \text{CER-GOD}  \\\\\hline
>     \text{OGB-Proteins} &72.67  & 71.34&71.42 &73.71 & 74.49& \bf74.81 \\\\\hline
> \end{matrix}$
>
> - We want to clarify that the datasets chosen in our initial paper are already from various **real-world** scenarios, such as **communication networks** (Email), **social networks** (Flickr), **citation networks** (Cora, CiteSeer), and **organic datasets** (Disney, Enron). Although these datasets are smaller than datasets like OGB-Proteins, they serve as standard benchmarks extensively used in classic and recent graph outlier detection literature [1,2,3]. Including them ensures a fair and comprehensive comparison with a wide range of baselines.
>
> **Reference:**
> > [1] Hezhe Qiao, Guansong Pang. Truncated Affinity Maximization: One-class Homophily Modeling for Graph Anomaly Detection. NeurIPS 2023.
>
> > [2] Jingyan Chen, Guanghui Zhu, Chunfeng Yuan, Yihua Huang. Boosting Graph Anomaly Detection with Adaptive Message Passing. ICLR 2024.
>
> > [3] Xiangyu Dong, Xingyi Zhang, Yanni Sun, Lei Chen, Mingxuan Yuan, Sibo Wang. SmoothGNN: Smoothing-aware GNN for Unsupervised Node Anomaly Detection. WWW 2025.
>
>
> **Response to W3:**  Thank you for your comment. Following your suggestion, we have included TAM [NeurIPS'23], SmoothGNN [WWW'25], and GADAM [ICLR'24] into our comparison. Below, we show the experimental results:
>
> $\begin{matrix}
>     \hline
>     &\text{Email} & \text{Cora} & \text{Disney} & \text{Flickr} & \text{CiteSeer} & \text{Enron} & \text{Reddit} \\\\
>     \hline
>     \text{TAM} & 30.45\pm0.01&55.55\pm0.37 &30.51\pm0.00 &\underline{65.19\pm0.86} & 46.75\pm1.40 & 44.75\pm0.03 &\underline{58.60\pm0.03}\\\\
>     \text{SmoothGNN} &51.09\pm15.37 & 62.72\pm8.14 & 54.40\pm8.64 & 50.26\pm4.26&52.82\pm1.99 & \underline{52.35\pm3.40} & 58.38\pm6.23\\\\
>     \text{GADAM} & \underline{68.12\pm2.39} &\bf92.62\pm0.35 & \underline{69.35\pm0.22} &61.46\pm0.22  & \bf93.91\pm0.13 & 33.91\pm0.47 & 58.44\pm0.26 \\\\
>     \text{CER-GOD} &\bf96.98\pm0.08 &\underline{92.09\pm1.26} &\bf 72.13\pm3.01&\bf67.08\pm0.16 &\underline{74.01\pm0.37} &\bf72.63\pm3.65 &\bf59.71\pm1.89\\\\\hline
> \end{matrix}$
>
> We can observe that CER-GOD achieves the best performance on 5 out of 7 datasets. In particular, on datasets such as Email and Enron, CER-GOD outperforms the strongest competitor, GADAM, by large margins (**+28.8\%** on Email, **+38.7\%** on Enron). On the other hand, we observe that GADAM achieves impressive performance on citation networks (Cora/CiteSeer). However, its performance collapses on other graph types (e.g., 33.91\% on Enron). In contrast, CER-GOD remains competitive on Cora (92.09\%) while maintaining SOTA performance across diverse domains (social, communication, and organic networks). We believe the evaluation across diverse benchmarks and the comparison with comprehensive SOTA baselines provide sufficient evidence to demonstrate the effectiveness of our method.

---

> ### Author Response · Authors · 2025-11-21
>
> **Response to W4:** We would like to address your concern via the following clarifications:
>
> - The fluctuations in Figure 3 arise because we **deliberately** set each hyperparameter over a very broad range (from $10^{-3}$ to $10^{3}$) in order to thoroughly evaluate their impact. As shown in the curves, the performance actually **remains stable across several orders of magnitude**, especially within certain ranges such as $[0.001, 1]$. Performance only significantly degrades when the hyperparameters become **extremely** small or large, which is theoretically expected since the corresponding loss term becomes negligible or overwhelmingly dominant. Thus, the model is actually robust when the hyperparameter is within a reasonable range.
>
> - Regarding the grid search in the paper, we followed the **standard evaluation protocol** in the graph outlier detection literature (e.g., ADA-GAD): While the training process is fully unsupervised, we utilized labels during the evaluation phase to report the maximum potential capacity of the method. As demonstrated in Figure 3, our model exhibits a broad "safe operating region" (e.g., $\alpha, \beta \in [0.01, 1]$), where performance remains consistently stable. Therefore, for a new dataset, users simply adopt a **fixed default configuration** (e.g., setting hyperparameters to 0.1) as a robust starting point. The stability analysis demonstrates that CER-GOD can achieve near-optimal performance by setting the hyperparameter values within the recommended range, which ensures its applicability.
>
> **Response to W5:**  To address your concern, we have shown the finetuned hyperparameters and searching protocol as follows:
> - We conducted a grid search for $\alpha, \beta, \gamma \in \\{10^{-3}, 10^{-2}, \dots, 10^{3}\\}$ (7 candidate values each). Based on preliminary experiments showing stability, we fixed $\gamma=1$ in all experiments to reduce computational cost. For each configuration, we follow the same training protocol and report AUC as the mean and standard deviation over 10 independent runs. Therefore, this results in a total of $49$ independent configurations evaluated for each dataset.
>
> - The optimal values identified through this grid search are listed below. We will include these details and the search grid specification in the Appendix for reproducibility.
>
> $\begin{matrix}\hline
>      & \text{Email} & \text{Cora} & \text{Disney} & \text{Flickr} & \text{CiteSeer} & \text{Enron} \\\\ \hline
>     \alpha & 0.001&0.1 &0.001 & 0.1& 0.001&0.001 \\\\ \hline
>     \beta & 0.001&10 &0.01 &0.001 &1 &1 \\\\\hline
>     \gamma & 1 & 1 & 1 & 1 & 1 & 1\\\\\hline
> \end{matrix}$
>
> - Additionally, we have provided saved .pth files of weights in the released code to ensure reproducibility.
>
> **We hope that the explanation and further analysis help to address your concerns.**

---

> > ### Comment · Reviewer_6mWp · 2025-11-27
> >
> > Thanks for the rebuttal. Although I appreciate the effort of the authors, I don't think it fully addresses my concern.
> >
> > First, I believe the authors should use the datasets specifically designed for graph outlier detection, as they are more and more used in the related area, such as [1-3], instead of using the node classification datasets, which may lack the statistical properties (imbalance, heterophily, etc). Therefore, I encourage the authors to follow the current trend of using the datasets, such as Amazon, T-Finance, Elliptic, DGraph-Fin, and T-Social in [4], to demonstrate their effectiveness.
> >
> > Second, I think it is also better to provide memory usage to show the efficiency.
> >
> > Third, the utilization of grid search to see the best performance can be cheating, as it is hard to decide when to stop for evaluation in real life. I believe the authors should set their hyperparameters based on other things. Here are possible solutions for that: 1. choosing the model with minimal loss on the training set. 2. use statistical information from graphs, such as density, to decide the hyperparameters for new datasets.
> >
> >
> > reference:
> >
> > 1. Hezhe Qiao, Guansong Pang. Truncated Affinity Maximization: One-class Homophily Modeling for Graph Anomaly Detection. NeurIPS 2023.
> >
> > 2. Nan Chen, Zemin Liu, Bryan Hooi, Bingsheng He, Rizal Fathony, Jun Hu, Jia Chen. Consistency Training with Learnable Data Augmentation for Graph Anomaly Detection with Limited Supervision. ICLR 2024.
> >
> > 3. Xiangyu Dong, Xingyi Zhang, Yanni Sun, Lei Chen, Mingxuan Yuan, Sibo Wang. SmoothGNN: Smoothing-aware GNN for Unsupervised Node Anomaly Detection. WWW 2025.
> >
> > 4. Jianheng Tang, Fengrui Hua, Ziqi Gao, Peilin Zhao, Jia Li. GADBench: Revisiting and Benchmarking Supervised Graph Anomaly Detection. NeurIPS 2023.

---

> ### Author Response · Authors · 2025-11-27
>
> **To FQ1:** We appreciate the reviewer’s follow-up, below are our clarifications:
> - We fully agree on the importance of using datasets that follow the current trend. This is exactly why, in our rebuttal, we have supplemented experiments on datasets such as **Reddit** (Response to Reviewer ErB5' W2) and **OGB-Proteins** (Response to W3), which are **from the papers [1][2][3][4] mentioned by the reviewer** and represent large-scale, real-world outlier detection scenarios. The results on these datasets demonstrated the effectiveness of CER-GOD.
> - Regarding the concern that our initial datasets lack statistical properties, we clarify that (1) Table 3 (Appendix F) explicitly reports their anomaly rates (ranging from **0.04\%** to **6\%**), which indicates they are clearly **highly imbalanced**. (2) Datasets such as Enron, Disney, etc, are widely recognized as organic graphs with **natural heterophily**.
> - The datasets used in our initial submission are not merely node classification datasets, they are also well-established benchmarks that have been used in many classical GOD works (e.g., BOND benchmark[NeurIPS'22], DOMINANT, AnomalyDAE, etc) and latest GOD works (e.g., BOURNE[ICDE'24], AD-GCL[AAAI'25], etc).  While we agree that including the benchmarks that the reviewer suggests can make our experiment more comprehensive (and **we did include**), we believe our initial choices do not diminish the value of our work.
> - Follow your advice, **we are now working on adding more datasets**. We will update the results as soon as possible. But we argue that the experiment now includes both **initial benchmarks** with the **newly added datasets**, such as Reddit and OGB-Proteins, suggested by the reviewer. We believe that our evaluation encompasses both standard benchmarks and the large-scale trend to ensure a comprehensive assessment.
>
> **To FQ2:**  Follow your suggestion, we here provide the quick comparison in terms of memory usage (MB) in each epoch between our method and several baselines:
>
> $\begin{matrix}
> \hline
>     &\text{DOMINANT} &\text{AnomalyDAE} &\text{CONAD} &\text{AS-GAE} &\text{ADA-GAD} &\text{BOURNE} &\text{AD-GCL} & \text{CER-GOD} \\\\\hline
> \text{Disney} &16.47 &16.62 &16.47 & 16.65& 64.14& 21.80 & 17.71 & 17.13\\\\
> \text{Email} &17.93 & 25.40&17.93 &22.08 &  68.35& 122.12 &158.98 &56.81\\\\
> \hline
> \end{matrix}$
>
> From the results, we can observe that the memory usage of CER-GOD is evidently **competitive** with other baselines, which we believe, together with the **running time comparison** (response to W1) and **anomaly detection performance** (Table 1 and the added SOTA baselines and datasets in the rebuttal), have demonstrated the effectiveness of CER-GOD.
>
> **To FQ3:** Below are our further clarifications:
> - First, we emphasize that **the grid search is not used for training, but rather for robustness checks**, and our training process is strictly unsupervised to ensure no label leakage during optimization. For each dataset, we (i) fix a **finite search space** around a default configuration (e.g., a logarithmic grid for
> $\alpha$ and $\beta$), (ii) train each candidate setting for a **fixed number of epochs**, and (iii) select the final configuration using only **training-time quantities**: (1) the final training objective loss (which is the same as solution 1 the reviewer mentioned), and (2) a simple non-collapse check on the cluster assignments (to avoid the trivial solution where all nodes fall into a single cluster).
> - Second, we realize that our description of the **parameter sensitivity analysis** may have caused confusion. In Figure 3, we evaluated the performance across a wide range of hyperparameter values using AUC. However, the only purpose here is to demonstrate robustness, not to select or tune hyperparameters. We will revise **Appendix G** to explicitly separate (i) the unsupervised hyperparameter selection protocol (based on the training loss and non-collapse) from (ii) the post-hoc sensitivity study (which uses labels purely for sensitivity analysis).
> - Third, as demonstrated in our parameter sensitivity analysis (**Figure 3**), the model exhibits a broad "Safe Operating Region." For example, within the range of $\alpha, \beta \in [0.01, 1]$, our AUC remains consistently above 90\%, which is still superior to the SOTA baselines, such as AD-GCL, ADA-GAD, SmoothGNN, TAM, etc. The experimental results show that CER-GOD does not rely on precise tuning to be effective. In real-world scenarios, users can safely adopt a default configuration (e.g., setting all hyperparameters to 0.1) without performing any dataset-specific search and still achieve competitive performance. We have also provided the recommended setting of hyperparameter values in the response to **W4\&W5**.
> We consider this clarifies that our strategy is principled, label-free, and practically usable, rather than a form of "cheating".
>
> **We look forward to your feedback and hope our response helps address your concern.**

---

> > ### Comment · Reviewer_6mWp · 2025-11-28
> >
> > I appreciate the timely rebuttal and the effort of the authors. I will increase my score and give my final comment when the authors provide satisfactory performance on real-world datasets.

---

> ### Author Response · Authors · 2025-12-04
>
> **Thank you for the follow-up feedback. We are glad to hear that your concerns about the efficiency and hyperparameter selection strategy can be addressed, and you're willing to increase the score**. We have supplemented the experimental results of the **Amazon** dataset following your suggestion. The performance is shown as follows:
>
> $\begin{matrix}
>     \hline
>     &\text{L1SUB} & \text{DEEPFD} &\text{GAT+ClusterAD} & \text{DOMINANT} & \text{AnomalyDAE} & \text{CONAD} &\text{AS-GAE}&\text{ADA-GAD}&\text{BOURNE} &\text{AD-GCL} &\text{SmoothGNN} &\text{GADAM}& \text{TAM}& \text{CER-GOD}\\\\\hline
>     \text{Amazon} & 47.53\pm0.02&50.00\pm0.00&65.43\pm9.60&50.36\pm0.59&39.80\pm6.70&48.94\pm1.56&48.96\pm0.52&50.86\pm1.10&75.01\pm7.15&24.94\pm3.32&49.96\pm0.08&57.15\pm1.50&79.87\pm0.16&\bf86.24\pm3.56\\\\\hline
> \end{matrix}$
>
> We can observe that CER-GOD consistently **outperforms** SOTA baselines on Amazon. Additionally, the datasets such as **Reddit** and **OGB-Proteins** that we added in the rebuttal **are all from the literature that you mentioned**, and CER-GOD achieved **satisfactory performance** on them as well. **We believe the added three datasets, together with the six datasets used in our initial submission, can well represent real-world, large-scale datasets to demonstrate the effectiveness of our CER-GOD.**

---

### Official Review · Reviewer_Q1xe · 2025-10-26

**Soundness:** 2
**Presentation:** 2
**Contribution:** 2
**Rating:** 2
**Confidence:** 3

**Summary:**

This paper introduces CER-GOD, a graph outlier detection framework designed to mitigate the "Homophily Trap" in graph convolutional networks (GCNs), where aggregating features from anomalous neighbors contaminates normal node representations. The method combines a self-discriminative masking spoiler that adaptively re-weights edges to reduce heterogeneous influences, guided by a learnable clustering layer that generates pseudo-labels without predefined thresholds. A diversity loss prevents class collapse during clustering. The overall objective balances reconstruction, clustering, distribution repulsion, and diversity terms. Experiments on six datasets show superior AUC performance over baselines. Contributions include: (1) analysis of the homophily issue with a masking mechanism, (2) threshold-free pseudo-labeling via clustering, and (3) a regularization to stabilize optimization.

**Strengths:**

The paper creatively combines edge re-weighting with unsupervised clustering for outlier detection, extending the "Homophily Trap" concept (He et al., 2024) into a joint optimization framework. The diversity loss is a novel tweak to address a specific failure mode in clustering under homophily constraints, potentially applicable beyond outliers.

**Weaknesses:**

1. The paper's core innovation—adaptive edge masking guided by binary clustering—feels incremental rather than transformative, building too closely on existing ideas without sufficient novelty. For instance, the masking spoiler resembles attention mechanisms in GAT or adaptive topology learning, but lacks a rigorous comparison showing why global MMD guidance (Eq. (8)) outperforms local feature-based weighting.
2. The "self-discriminative" claim is undersold: it primarily enforces intra-cluster aggregation via pseudo-labels, which is akin to DEC (Guo et al., 2017) but without justifying why two clusters suffice for diverse anomaly types (e.g., point vs. structural outliers).
3. Proposition 1, while neat, is a straightforward extension of over-squashing bounds (Topping et al., 2022) and does not uniquely motivate the framework—empirical validation in Fig. 1 is limited to one dataset (Email) and single-layer GCN, ignoring multi-layer or heterophilic graphs.

**Questions:**

1.	Can the authors clearly explain how the proposed masking differs mathematically from the attention coefficients in GAT or the relation-strength function in ADA-GAD?
2.	How sensitive is the model to the initial K-Means clustering used for pseudo-label initialization?
3.	Does the method scale to larger graphs (e.g., >100k nodes)? Have the authors considered mini-batch or approximate MMD computation?
4.	Can the authors provide quantitative evidence (e.g., correlation coefficients) showing that the learned edge weights actually correlate with homophily or anomaly boundaries?

---

> ### Author Response · Authors · 2025-11-21
>
> **We thank the reviewer for the comments. Please find our responses to your concerns below.**
>
> **Response to W1:** We appreciate the reviewer's comment. However, we respectfully clarify that our masking spoiler is not an incremental innovation. We have discussed how the masking spoiler differs fundamentally from local attention (such as GAT), as outlined in **Section 3**. Here, we would like to address your concern from the following three perspectives:
>
> - **Local Weighting (e.g., GAT, Adaptive Topology):** Existing methods typically re-weight edges based on local feature similarity or smoothness. As noted in Section 1, these methods fail under the "Homophily Trap" because anomalous nodes often mimic the features of their neighbors ("camouflaged anomalies"). In such cases, local attention mechanisms erroneously assign high weights to these edges, exacerbating contamination.
> - **CER-GOD:** In contrast, our masking spoiler is not a function of local feature pairs ($z_i, z_j$). Instead, it acts as a learnable topology filter and is explicitly optimized to maximize the distributional distance (MMD) between the normal and anomalous clusters. It learns to suppress an edge not because the nodes are dissimilar, but because the edge's existence reduces the global discriminability of the two clusters. This allows CER-GOD to identify and sever connections that local mechanisms would mistakenly preserve.
> - **Empirical Evidence:** We clarify that we have provided a direct ablation-style comparison in Table 1, i.e., the comparison with the baseline "GAT+ClusterAD" that represents the "Local feature-based weighting + Clustering" approach. We can observe that CER-GOD consistently outperforms GAT+ClusterAD by a significant margin (e.g., +30\% on Email, +15\% on Flickr). Furthermore, the embedding visualization in Figure 4 shows that GAT-based baselines (e.g., DOMINANT, GAT+ClusterAD) still suffer from overlap (Homophily Trap), whereas CER-GOD achieves a clearer separation. These empirical results quantify precisely why global MMD guidance is superior to local weighting for anomaly detection.
>
> **Response to W2:** We would like to address your concern by clarifying the definition of "self-discriminative" and why we adopt two clusters:
>
> - **"Self-Discriminative" vs. DEC:** We clarify that our framework differs fundamentally from DEC. DEC focuses on optimizing embeddings within the feature space via the KL-divergence. However, clustering is only one component of our method. Our "self-discriminative" mechanism operates actively on the graph topology. The core innovation is the Masking Spoiler, which learns to discriminate between beneficial and harmful neighbors by re-weighting edges based on feedback from the clustering. This creates a more sophisticated and synergistic loop: the clustering-based detector produces pseudo-labels that are fed as signals to the masking spoiler. These pseudo-labels define the $\mathcal{D}\_{pos}$ and $\mathcal{D}\_{neg}$ sets, which then optimizes the learnable mask $M$ to suppress the edges that cause cluster contamination (via MMD loss). Subsequently, the encoder learns more discriminative embeddings in return for refining the clustering. Thus, "self-discriminative" refers to the model's ability to autonomously purify its own input structure, a capability that DEC does not possess.
>
> - **Why Two Clusters:** In this paper, we followed the classical one-class classification setting as adopted by previous unsupervised graph outlier detection works [1,2,3]: The "anomalous candidate cluster" does not assume all anomalies are homogeneous, but instead treats anomalous patterns (e.g., point/structural/attribute anomalies) as different manifestations of deviating from the manifold with respect to the normal pattern. Therefore, we adopt $c=2$ not to imply that all anomalies are identical, but because our edge reweighting mechanism unifies the treatment of diverse anomaly types. Our masking spoiler model is trained to identify these heterogeneous connections and significantly mitigate their impact, thereby maximizing global separability. By weakening the aggregation effects originating from these neighbours, the model effectively prevents feature contamination, keeping point outliers and structural outliers distant from the compact normal manifold in the latent space. The empirical evidence can refer to the embedding visualization in Figures 4 and 9, CER-GOD exhibits a significantly clearer separation between normal nodes and diverse anomalies compared to baselines.
>
> **References:**
> > [1] Deep one-class classification. ICML, 2018.
>
> > [2] Deep anomaly detection on attributed networks. ICDM, 2019.
>
> > [3] Unsupervised deep subgraph anomaly detection. ICDM, 2022.

---

> > ### Author Response · Authors · 2025-11-21
> >
> > **Response to W3:** We would like to address your concern via the following clarifications:
> >
> > - The **motivation for Proposition 1** was not to claim a new mathematical bound, but to formally connect the principles of information propagation (as in over-squashing) specifically to the context of the "Homophily Trap". The proposition quantifies the intuition from Figure 1: the contaminating influence of an anomalous node decays exponentially with distance $r$. This mathematically justifies why 1-hop neighbors (where $r$ is small) are the primary source of contamination, and why the proposed mask strategy that directly targets these connections is a necessary solution.
> > - For the **empirical validation** in Figure 1, we clarify that we **deliberately** utilized a single-layer GNN without any linear projection operation to isolate the root cause of the "Homophily Trap" problem. Multi-layer GCNs compound aggregation, which would obscure whether the contamination originates from the immediate neighbor or propagated noise. Following your suggestion, we have extended this analysis to two synthetic (coming from [1]) and two real-world datasets (Cora and Flickr) in Fig. 8 of the revised paper. We also calculate the related statistical information (Average MMD distances between standard Gaussian distribution and anomalous nodes, 1-hop normal nodes, and multi-hop normal nodes, respectively) to evaluate the influence:
> >
> > \begin{matrix}
> > \hline
> >     &\text{Avg MMD of Anomalous Nodes} & \text{Avg MMD of 1-Hop Normal Nodes} & \text{Avg MMD of Multi-Hop Normal Nodes} \\\\\hline
> > \text{Cora} & 8.532793& 7.048123&6.98577 \\\\
> > \text{Flickr} & 6.421008& 6.618733& 4.93772\\\\\hline
> > \end{matrix}
> > The results provide empirical evidence for the universality of the "Homophily Trap", where the 1-hop normal nodes exhibit a more significant distributional shift compared to multi-hop normal nodes, and align closely with the deviation of anomalous nodes. This quantitatively proves that proximity to anomalies severely contaminates the embeddings of normal neighbors.
> > >[1] Unsupervised deep subgraph anomaly detection. 2022 IEEE International Conference on Data Mining (ICDM). IEEE, 2022.
> >
> > - We further clarify that even in heterophilic graphs, the "Homophily Trap" persists because standard GCNs will indiscriminately aggregate these dissimilar features. Our analysis holds because anomalies often act as "noisy heterophily", which needs to be distinguished from "informative heterophily".
> >
> > **Response to Q1:** Thank you for your concern, and we would like to address your concern by comparing with GAT and ADA-GAD:
> >
> > -  GAT's attention coefficient is a dynamic function of node features:
> > $\alpha\_{ij} = \text{Softmax}(\text{LeakyReLU}(\mathbf{a}^T [W\mathbf{h}\_i || W\mathbf{h}\_j]))$.
> > This implies that edge weights are determined solely by local feature similarity. Under the "Homophily Trap", if an anomalous node mimics its neighbor's features, GAT inevitably assigns a high $\alpha_{ij}$, which will propagate noise.
> >
> > - The relation-strength function of ADA-GAD is formulated as $S\_{ano}(\mathcal{G}) = E\_{high}(\mathcal{G}, \boldsymbol{D}) = \frac{\boldsymbol{D}^T \boldsymbol{L} \boldsymbol{D}}{\boldsymbol{D}^T \boldsymbol{D}},$
> > This essentially enforces graph smoothness, which masks edges that connect dissimilar nodes. However, this approach fails when anomalies are structurally connected but have similar features, as it relies on a predefined smoothness prior rather than a task-specific objective.
> >
> > - In CER-GOD, the mask $M \in \mathbb{R}^{N \times N}$ is defined as a set of independent learnable parameters ($\tilde{A} = \sigma(M) \odot A$), rather than a dynamic function of local feature pairs or a fixed smoothing heuristic. Its optimization is explicitly driven by the distribution repulsion loss, which trains the mask to suppress edges that degrade the distributional separability between normal and anomalous candidate clusters. This makes the proposed masking strategy fundamentally different from GAT and ADA-GAD.
> >
> > **Response to Q2:**  As described in Algorithm 1, $K$-Means is performed only for the initialization of the cluster centroids $\mu$. Even with random initialization, the GCN backbone performs Laplacian smoothing to aggregate neighborhood information, which allows $K$-Means to operate on structured, semantically meaningful representations rather than random noise. Additionally, the cluster centroid $\mu$ is a learnable parameter that is iteratively updated during training through the clustering loss. This allows the model to dynamically correct suboptimal deviations from the initialization.
> >
> > In our initial submission, we have verified this by reporting the mean and standard deviation over 10 independent runs (random $K$-Means initialization) in **Table 1**. The low standard deviations (e.g., $\pm 0.08\%$ on Email, $\pm 0.16\%$ on Flickr) serve as empirical evidence that the model is stable and robust with random initialization of $K$-means.

---

> > ### Comment · Reviewer_Q1xe · 2025-11-28
> >
> > Authors have addressed all of my concerns. I would like to raise the score to 6.

---

> ### Author Response · Authors · 2025-11-21
>
> **Response to Q3:** To address your concern, we have scaled CER-GOD to large graphs ($\\gt$100k nodes) through a mini-batch adaptation. Specifically:
>
> - **Mini-batch Strategy:** To handle large-scale graphs, we implemented a mini-batch training scheme. We clarify that this mini-batch approach effectively acts as an approximate MMD computation. While exact MMD requires global statistics, our implementation approximates the distributional distance using batch-level statistics. A challenge here is that standard random batching may yield batches without anomalous nodes. We address this by computing the repulsion loss only for batches containing both candidate types. In practice, using a larger batch size will ensure stable gradient updates.
>
> - **Empirical Scalability Analysis:** We evaluated CER-GOD on the OGB-Proteins dataset, which contains **132,534 nodes** and **79,164,284 edges**. As shown in the table below (note that we reported the performance of all baseline methods directly from[1]), CER-GOD scales successfully and outperforms several state-of-the-art baselines in comparison.
> \begin{matrix}
>     \hline
>     &\text{DOMINANT} &\text{ComGA} & \text{CoLA}& \text{SL-GAD} & \text{TAM} & \text{CER-GOD}  \\\\\hline
>     \text{OGB-Proteins} &72.67  & 71.34&71.42 &73.71 & 74.49& \bf74.81 \\\\\hline
> \end{matrix}
>
> > [1] Truncated Affinity Maximization: One-class Homophily Modeling for Graph Anomaly Detection. NeurIPS 2023.
>
> **Response:** We appreciate the reviewer's suggestion. We here provided both the qualitative visualization (as included in the initial submission) and the newly added quantitative statistics to demonstrate this correlation:
> - **Qualitative Verification:** As shown in Figure 11 (Appendix), we visualized the masking strength on the Flickr dataset. Elements corresponding to normal-anomalous node pairs (the cause of the "homophily trap") are highlighted in black frames, and it is evident that their connections are significantly weakened (shown in blue/white) in the masked topology.
>
> - **Quantitative Evidence:** To strictly quantify the correlation between learned edge weights and homophily labels, we calculated the average mask weight for Intra-class edges (Normal-Normal) versus Inter-class edges (Normal-Anomaly) on the Flickr dataset. The results are:
> - Intra-class Edges (Homophilous): 0.5769
> - Inter-class Edges (Heterophilous): 0.1311
>
> These results provide strong evidence to demonstrate that the learned edge weights actually correlate with homophily.
>
> **We hope that the explanation and further analysis help to address your concerns.**

---

> ### Author Response · Authors · 2025-12-04
>
> **Thank you for the positive feedback and your decision to raise the score.** We are glad to hear that our rebuttal can address your concerns, particularly regarding **several important aspects** such as **novelty**, **the clarification on "self-discriminative"**, and **the scalability issue**, etc. We appreciate the insightful suggestions you provided and **have incorporated relevant content added during the rebuttal phase into the revised manuscript**.

---

### Official Review · Reviewer_ErB5 · 2025-10-30

**Soundness:** 2
**Presentation:** 3
**Contribution:** 2
**Rating:** 4
**Confidence:** 5

**Summary:**

This paper proposes a learnable clustering-guided edge reweighting framework that jointly optimizes a self-discriminative masking module and an adaptive clustering-based outlier detector. The clustering process generates pseudo labels in an unsupervised manner, which are then utilized to guide the training process.

**Strengths:**

(1) The method is straightforward to understand, but the paper’s writing quality needs improvement.  am particularly confused about the role of the pseudo labels.

(2) The proposed method achieves the best performance across six small-scale graph anomaly detection  datasets.

**Weaknesses:**

(1) The authors should provide a clear technical definition or explanation for the newly introduced concepts “multi-hop away from anomalies” and “1-hop away from anomalies.” For instance, a 1-hop-away node should refer to a normal node that is directly connected to at least one anomalous node.
Additionally, the description of “data distribution” in the framework is misleading — it should be clarified as the anomaly score distribution or MMD score distribution if you use the histogram. Moreover, the current explanation of the framework implies that only two clusters are considered in the diversity loss. The role of the pseudo-labeling process is also not clearly reflected in the framework diagram, and the legend is missing. The type of latent embedding *z* is not consistent with the main paper. The authors should revise the figure and explanation to make these components explicit.

(2)Most of the benchmark datasets used are relatively small, making the experimental results less reliable—especially on Enron, which contains only five anomalies. I suggest that the authors include larger-scale datasets, covering both more injected or real-world data with real anomalies, to better demonstrate the robustness and effectiveness of the proposed method.

(3) The approach appears to focus primarily on the reconstruction process and the learning of discriminating representations.  The unsupervised clustering module is learnable, and the inference relies on the distance to the centroids. In this case, it is unclear why a reconstruction component is still necessary. Why not directly apply clustering to the graph representations learned by the GCN encoder? Moreover, the ablation study is not sufficiently comprehensive—additional variants, like directly applying the learnable clustering, should be included for a more thorough evaluation

(4)Determining the appropriate number of clusters for each dataset is challenging. How do the authors decide on the cluster count in practice? Additionally, the reconstruction process is computationally complex, and the inclusion of clustering further increases the overall computational cost. Besides, the performance with varying the number of clusters should be provided.

(5) As you mention in lines 244-247, ”Then we first designate the cluster containing a relatively larger number of samples as the normal cluster, and temporarily treat all nodes within it as normal candidates. Conversely, the remaining clusters are considered the anomalous candidate cluster“.  How many large clusters are treated as normal nodes? In real GAD datasets, the data distribution often consists of multiple normal clusters with relatively large sample sizes, along with multiple anomalous clusters and some isolated outlier points.

**Questions:**

See above **Weaknesses**

---

> ### Author Response · Authors · 2025-11-21
>
> **We thank the reviewer for the comments. Please find our responses to your concerns below.**
>
> **Response to W1:** We have revised the related content to address all these points:
> - We have supplemented the technical definition of '$k$-hop away from anomalies' in the Introduction section as follows: ``Formally, a "normal node 1-hop away from anomalies" denotes a normal node with a direct edge (1-hop distance) to at least one anomalous node, while a "normal node multi-hop away from anomalies" is a normal node at a distance of two or more hops from its nearest anomalous neighbor.''
> - We acknowledge that the term "Data Distribution" in Figure 2 was imprecise. While this module measures distributional discrepancy (via MMD) against a reference standard Gaussian distribution, the operation is performed specifically on the learned latent embeddings rather than the raw input data or the final anomaly scores. Therefore, to accurately reflect this and avoid confusion, we have corrected the term to "Latent MMD Score Distribution" and added a detailed clarification in the revised manuscript.
> - Regarding the Diversity Loss, we clarify that the mathematical formulation is actually generic and capable of handling any number of clusters $c\ge2$. While the framework supports multi-cluster differentiation, we configured $c=2$ in our experiments to ensure robust unsupervised detection. Please refer to our response to W4 for the detailed rationale and empirical validation of this design.
> - We have updated the pseudo-labeling process in the framework. Each instance's prediction is now colored, allowing the model to partition nodes into normal and anomalous candidate groups according to their pseudo-labels. Additionally, we added a legend to enhance clarity. These groups are subsequently used to optimize the distribution repulsion loss.
> - We use $\mathbf{z}_i$ to denote the embedding of the
> $i$-th sample and $\mathbf{Z}$ to denote the complete embedding matrix. We have corrected the notation from $\mathcal{D}=\\{\mathbf{Z}\_{i}^{(0)}\\}\_{i=1}^{N}$ to $\mathcal{D}=\\{\mathbf{z}\_{i}^{(0)}\\}\_{i=1}^{N}$ for consistency (in line 204).
>
> **Response to W2:** We have supplemented a new experiment conducted on the Reddit dataset, which contains **366** real-world outliers. The results are shown as follows:
>
> $\begin{matrix}
> \hline
> &\\text{L1SUB} & \text{DEEPFD} &\text{GAT+ClusterAD} & \text{DOMINANT} & \text{AnomalyDAE} & \text{CONAD} &\text{AS-GAE}&\text{ADA-GAD}&\text{BOURNE} &\text{AD-GCL} & \text{CER-GOD} \\\\\hline
> \text{Reddit} &56.94\pm0.00 & 51.67\pm0.00&54.78\pm3.18 &55.88\pm0.43 & 48.38\pm2.97 & 56.12\pm0.03 &49.35\pm5.01 &56.17\pm0.10 & 57.48\pm2.28&53.94\pm0.79 &\bf59.71\pm1.89\\\\
> \hline
> \end{matrix}$
>
> We can observe that CER-GOD consistently outperforms the SOTA baselines, which demonstrates the effectiveness of CER-GOD in handling a large amount of anomalies. Besides, we have also conducted experiments to verify the scalability of CER-GOD on large-scale datasets (e.g., $\gt$100k nodes). Please refer to the response to **Reviewer Q1xe's Q3** for details.
>
> **Response to W3:** While the clustering module focuses on separability and structure, the reconstruction module plays a critical complementary role for two reasons:
> - Attribute reconstruction process forces the learned embeddings to **preserve** rich semantic information. Without reconstruction, the model is optimized solely for clustering. This potentially leads to "feature collapse," where embeddings become overly simplified, separating only clusters but losing the original node attribute information for identifying anomalies.
> - The structure reconstruction is another crucial part of the reconstruction component, which ensures the learned embeddings **respect** the original graph topology.
>
> To empirically demonstrate the effect of the reconstruction component, we follow your suggestion to add a "w/o Reconstruction" variant, which applies the learnable clustering directly to the GCN encoder, to our ablation study:
>
> $\begin{matrix}
>     \hline
>     &\text{Email} &\text{Cora} &\text{Flickr}\\\\\hline
>     \text{w/o Reconstruction} & 50.80\pm3.10& 52.20\pm14.33 &49.24\pm2.35 \\\\
>     \text{Ours} &\bf96.98\pm0.08 &\bf92.09\pm1.26 &\bf67.08\pm0.16 \\\\
>     \hline
> \end{matrix}$
>
> The results show that removing the reconstruction component leads to a significant performance drop, which suggests the necessity of this component in our method.
>
> Regarding the results of directly applying the learnable clustering, we have already conducted this experiment in our initial submission, which is referred to as 'w/o SD-MS' in **Table 2**. The results are shown below for your convenience:
>
> $\begin{matrix}
> \hline
>     \text{Methods/Datasets} & \text{Email}
>  & \text{Cora} & \text{Flickr}\\\\ \hline
> \text{\textbf{w/o} SD-MS} &87.05\pm4.66 & 78.84\pm2.68&58.40\pm6.61 \\\\
> \text{Ours} &\bf96.98\pm0.08 &\bf92.09\pm1.26 &\bf67.08\pm0.16 \\\\\hline
> \end{matrix}$

---

> ### Author Response · Authors · 2025-11-21
>
> **Response to W4:** We would like to address your concerns via the following three aspects:
>
> - In this paper, we focus on the **unsupervised outlier detection** task. In a fully unsupervised setting, we lack labels to map multiple learned clusters to the binary ground truth. For example, if we set $c=5$, determining which specific subsets of clusters represent "normal" and which represent "anomalous" requires additional heuristics (e.g., "the smallest 3 clusters are anomalies"), which introduces significant instability and potential bias. Therefore, we followed the **standard one-class classification setting** and set the number of clusters as $c=2$, as many previous works [1,2,3,4,5,6] did: The "anomalous candidate cluster" does not assume all anomalies are homogeneous, but instead treats anomalous patterns (e.g., point/structural/attribute anomalies) as different manifestations of deviating from the manifold with respect to the normal pattern.
> - Regarding computational cost, as analyzed in Appendix C, the reconstruction process is indeed a major time-consuming component in our method, and we have demonstrated its necessity in response to the W3. Compared to the reconstruction component, the clustering overhead is **linear** to $N$ and proportional to the small $c$ (here $c=2$). Thus, adding clustering does not change the overall asymptotic time complexity and adds only a modest overhead in practice.
> - Following your suggestion, we conducted an analysis varying the number of clusters $c$ from 2 to 5 on Flickr (the minority cluster is treated as the anomaly). In our test, we found that increasing $c$ did not yield performance gains and, instead, led to consistent performance degradation (e.g., AUCs all dropped by more than 5\%). This empirical finding suggests that splitting the normal class into multiple sub-clusters without supervision introduces fragmentation rather than clarity, validating that our binary objective ($c=2$) aligns best with the goal of unsupervised anomaly detection.
>
> **References:**
>
> > [1] Deep one-class classification. ICML, 2018.
>
> > [2] Deep anomaly detection on attributed networks. ICDM, 2019.
>
> > [3] Unsupervised deep subgraph anomaly detection. ICDM, 2022.
>
> > [4] Truncated affinity maximization: One-class homophily modeling for graph anomaly detection. NeurIPS, 2023.
>
> > [5] ADA-GAD: Anomaly denoised autoencoders for graph anomaly detection. AAAI, 2024.
>
> > [6] Revisiting graph contrastive learning on anomaly detection: A structural imbalance perspective. AAAI, 2025.
>
> **Response to W5:** As explained in W4, our framework is theoretically flexible. In scenarios with known multi-modal normal distributions, the top $k$ largest clusters can be designated as the "Normal Candidate Set." However, to maintain a robust, threshold-free framework without prior knowledge of the number of semantic classes, we rely on the unsupervised assumption that the majority belongs to the normal distribution ($c=2$). Our supplemented analysis of varying cluster numbers in W4 further confirms this is the optimal choice for these benchmarks.
>
> - **Why this works:** While the input data distribution may be multi-modal, the proposed Self-Discriminative Masking Spoiler (SD-MS) can minimize the influence of dissimilar neighbors by adaptively re-weighting edges. This process tends to align the latent embeddings of disparate "normal" sub-groups, which effectively compresses multi-modal normal patterns into a more compact, unified representation. Empirical evidence can be found in our latent embedding visualizations in Figures 4 and 9 of our paper, which demonstrate that our method effectively aligns multi-modal normal data while distinctly separating them from anomalies.
>
> **We hope that the explanation and further analysis help to address your concerns.**

---

> > ### Comment · Reviewer_ErB5 · 2025-11-27
> >
> > Thanks for the detailed rebuttal. The response has addressed most of my concerns, especially regarding the role of clustering and the choice of hyperparameter $c$ in the clustering process. I initially intended to raise the score to positive, but I noticed that the authors adopted grid search for the two crucial hyperparameters. In an unsupervised setting, we do not have a validation set, so grid search is not applicable. For the hyperparameters $\alpha$ and $\beta$, I would like to offer the following suggestions:
> >
> > (1) Use consistent or default values across all datasets. If this is not feasible, please provide a very small set of candidate values and identify some practical priors (eg, the scale or other statistical information) that can guide hyperparameter selection for new datasets. This is crucial for real-world deployment for unsupervised GAD.
> >
> > (2) Monitoring the minimization of the training loss or checking whether the training loss converges as the number of epochs increases can be used to determine the performance of GER-GOD.

---

> ### Author Response · Authors · 2025-11-27
>
> **Response to follow-up concern:** Thank you for your response, we are glad to hear that our rebuttal has addressed your initial concerns and your intention of raising the score. Regarding the new concern about the grid search strategy used in our experiment, we would like to make further clarifications:
>
> - **Regarding your Suggestion (1):** we actually have done the analysis and provided the **optimal and recommended settings of hyperparameter values** for real-world development (**refer to our response to Reviewer 6mWp's W4\&W5**). Additionally, as demonstrated in our parameter sensitivity analysis (Figure 3), the model exhibits a broad "safe operating region". For example, within the range of $\alpha,\beta\in [0.01, 1]$, the AUC remains consistently high and still superior to strong SOTA baselines such as AD-GCL, ADA-GAD, SmoothGNN, TAM, etc. These results show that CER-GOD **does not rely on precise hyperparameter tuning to be effective**.
>
> - **Regarding your Suggestion (2):** **First**, we emphasize that our training process is strictly unsupervised to ensure no label leakage during optimization. For each dataset, we (i) fix a **finite search space** around a default configuration (e.g., a logarithmic grid for $\alpha$ and $\beta$), (ii) train each candidate setting for a fixed number of epochs, and (iii) select the final configuration using only **training-time quantities**: 1) **the final training objective loss** (which is in line with **Suggestion (2)** raised by the reviewer), and 2) a simple **non-collapse check** on the cluster assignments (to avoid the trivial solution where all nodes fall into a single cluster). **Second**, we realize that our description of the **parameter sensitivity analysis** may have caused confusion. In Figure 3, we evaluated the performance across a wide range of hyperparameter values using AUC. However, the only purpose of this analysis is to demonstrate **robustness** of our method, not to select or tune hyperparameters. We will revise **Appendix G** to explicitly separate (i) the unsupervised hyperparameter selection protocol (based on the training loss and non-collapse check) from (ii) the post-hoc sensitivity study (which calculates AUC **purely** for sensitivity analysis).
>
> **We hope our clarifications help address your concern, and we look forward to your feedback.**

---

> > ### Comment · Reviewer_ErB5 · 2025-11-28
> >
> > Thank you for the detailed explanation regarding the hyperparameter settings. I am willing to raise my score to a positive rating and would like to suggest that the authors: (1) provide ablation results across all datasets including the newly added datasets, only **three datasets** are analyzed in the main paper and (2) report the **candidate values** for each hyperparameter along with a detailed analysis in the appendix, particularly how the parameters should be **selected** when encountering a new dataset. This is especially critical in real-world inference scenarios where no labels are available. Regarding the hyperparameters, moderate fluctuations are acceptable, but they should not **collapse**; thus, the authors need to clearly determine an appropriate and stable range.

---

> > > ### Author Response · Authors · 2025-12-04
> > >
> > > **Thank you for the follow-up feedback and your decision to raise the score**. We are glad to hear that our response addressed your concern regarding hyperparameter selection. For Suggestion (1), we will supplement more ablation results. For Suggestion (2), we have revised our paper to provide detailed settings for each hyperparameter value and guidance for implementing new datasets (in **Appendix G**). Additionally, we have demonstrated that our method is stable across a wide range of hyperparameter values and provided an appropriate and stable range (in **Section 4.3**).

---

### Official Review · Reviewer_A5bT · 2025-10-31

**Soundness:** 3
**Presentation:** 4
**Contribution:** 4
**Rating:** 8
**Confidence:** 4

**Summary:**

This paper proposes CER-GOD, a novel graph outlier detection framework to address the '' Homophily Trap'', which is a critical issue where graph convolutional operations blur the feature representations of normal and anomalous nodes that are neighbours. The proposed method synergistically integrates two main components: a self-discriminative masking spoiler that learns to reweight graph edges to suppress contaminating information flow from heterogeneous neighbours, and a clustering-based outlier detector that generates unsupervised pseudo-labels to guide this reweighting process. To ensure stable training and prevent clustering collapse, a diversity loss is introduced as a regularization term. Extensive experiments on multiple benchmark datasets demonstrate that CER-GOD significantly outperforms a wide range of state-of-the-art baselines.

**Strengths:**

1. The paper addresses a critical and well-articulated problem in GNN-based anomaly detection: the "Homophily Trap". The authors provided clear motivation, supported by insightful empirical analysis (Figure 1), highlighting how neighbourhood aggregation can contaminate node representations and fundamentally hinder outlier identification.
2. The proposed CER-GOD framework is methodologically sound and the idea is novel. The main innovation lies in the synergistic joint optimization of a self-discriminative masking spoiler and a clustering-based detector. This design creates a powerful feedback loop where pseudo-labels from clustering guide the edge reweighting, and the refined graph structure, in turn, yields more discriminative embeddings for improved clustering.
3. The comparison against a comprehensive set of baselines fully validates the effectiveness of CER-GOD. Additionally, the authors provided convincing qualitative evidence, such as t-SNE and mask visualizations, to further enhance the  persuasiveness and interpretability of the approach.
4. The authors provided the implementation code of the proposed method, which increases the reproducibility.

**Weaknesses:**

1. The choice of the Chebyshev distance for the MMD kernel calculation should be elaborated. While the intuition is provided, the paper would be strengthened by an empirical comparison against a more conventional Euclidean-based RBF kernel to justify this specific design.

2. The diversity loss $l_{diversity}$ introduces a crucial hyperparameter $\epsilon$ to control the minimum proportion of samples per cluster. However, there is no discussion regarding the setting of $\epsilon$ or any corresponding parameter sensitivity analysis.

3. A comparison with existing graph rewriting baselines is missing, which would help validate the method's effectiveness.

**Questions:**

1. Please also include a parameter sensitivity analysis examining how the distribution repulsion loss performs when applied to different graph convolutional layers.
2. The paper distinguishes the edge reweighting mechanism from graph rewriting methods. Could the authors further elaborate on the fundamental advantages of such learnable masks over heuristic-based hard edge removal or addition? A deeper discussion would be valuable.
3. The framework relies on a feedback loop where pseudo-labels guide the mask optimization. In the early training phase, these pseudo-labels might be noisy, which potentially leads the model to a suboptimal solution. How does the model ensure stability during the training phase? Is there a warm-up phase, or does the joint optimization naturally navigate towards a good solution?

---

> ### Author Response · Authors · 2025-11-21
>
> **We thank the reviewer for the comments. Please find our responses to your concerns below.**
>
> **Response to W1:** Follow your suggestion, we have conducted an empirical comparison between the Chebyshev-based kernel and the RBF kernel. The results are summarized below:
>
> \begin{matrix}
> \hline
> \text{Datasets} &\text{E-mail} & \text{Cora} & \text{Disney}\\\\
> \hline
>  \text{RBF kernel} &93.65\pm7.38 & 81.89\pm0.11 &68.69\pm8.05 \\\\
>  \text{Chebyshev-based kernel} & \bf96.98\pm0.08 &  \bf92.09\pm1.26 & \bf72.13\pm3.01\\\\
> \hline
> \end{matrix}
>
> The results in the table show that the Chebyshev-based kernel consistently outperforms the conventional RBF kernel across all three datasets. We attribute this improvement to the geometric properties of the metrics: the RBF kernel aggregates differences across all dimensions, which potentially dilutes anomalous deviations through a "smoothing" effect in high-dimensional spaces, while the Chebyshev kernel focuses exclusively on the maximum discrepancy along any single dimension. Given that anomalies generally exhibit as deviations in specific feature subsets rather than uniform global shifts, the Chebyshev kernel property enables the model to capture critical outlier patterns more effectively.
>
> **Response to W2:** We have included a detailed sensitivity analysis of the diversity loss hyperparameter $\epsilon$ in the revised manuscript, and the experimental results are summarized as follows:
>
> \begin{matrix}
> \hline
>     &0.1& 0.2& 0.3& 0.4& 0.45& 0.5 \\\\
>     \hline
> \text{Email} & 96.96\pm0.35&95.77\pm3.86 & 96.93\pm0.33&\bf97.06\pm0.14&96.98\pm0.08 &96.83\pm0.26\\\\
> \text{Disney} &62.62\pm3.35&66.67\pm2.38&61.21\pm0.85&69.02\pm1.91&\bf72.13\pm3.01 & 70.72\pm0.47\\\\
> \text{CiteSeer} & 66.20\pm11.77& 57.11\pm11.60  & 70.71\pm1.53 & \bf74.76\pm0.13&  74.01\pm0.37 &70.74\pm5.79\\\\
> \hline
> \end{matrix}
>
> We can observe that excessively small values may be overly permissive and further enable cluster collapse, thereby compromising performance across all datasets. Instead, $\epsilon=0.5$ denotes the strictest setting, which may slightly hurt performance because forcing an exact 50:50 balance may be too rigid for the natural structure of the data.
>
> **Response to W3:** Thank you for your comment. We supplement a graph rewriting baseline, TAM (NeurIPS'23) [1], to strengthen the competitiveness of the proposed method. The results are shown as follows:
>
> \begin{matrix}
>     \hline
>     &\text{Email} & \text{Cora} & \text{Disney} & \text{Flickr} & \text{CiteSeer} & \text{Enron} & \text{Reddit} \\\\
>     \hline
>     \text{TAM} & 30.45\pm0.01&55.55\pm0.37 &30.51\pm0.00 &65.19\pm0.86 & 46.75\pm1.40 & 44.75\pm0.03 &58.60\pm0.03\\\\
>     \text{CER-GOD} &\bf96.98\pm0.08 &\bf92.09\pm1.26 &\bf 72.13\pm3.01&\bf67.08\pm0.16 &\bf74.01\pm0.37 &\bf72.63\pm3.65 &\bf59.71\pm1.89\\\\\hline
> \end{matrix}
> CER-GOD demonstrates superior performance over TAM across all evaluated datasets, with improvements ranging from 21.88 points (Disney) to 66.53 points (Email), confirming its effectiveness across diverse graph structures and anomaly patterns.
> > [1] Hezhe Qiao, Guansong Pang. Truncated Affinity Maximization: One-class Homophily Modeling for Graph Anomaly Detection. NeurIPS 2023.
>
> **Response to Q1:** We have added the parameter sensitivity of using different layers within 5 layers of the graph encoder to calculate the distribution repulsion loss. The results are shown as follows:
>
> \begin{matrix}
> \hline
>     \text{Datasets/Layer} & 1 & 2& 3 &4 &5 \\\\\hline
>     \text{Disney} &\bf72.13\pm3.01 &70.72\pm0.47 &66.62\pm2.39 & 69.02\pm1.91 &61.21\pm0.85\\\\
>     \text{Enron}&\bf73.15\pm0.98 &73.00\pm0.05 &71.12\pm7.53 &72.70\pm3.93 &72.63\pm3.65  \\\\
>   \hline
> \end{matrix}
> It can be seen that performance consistently decreases as the layer depth increases from 1 to 5, with the first layer yielding optimal results.

---

> ### Author Response · Authors · 2025-11-21
>
> **Response to Q2:** The fundamental advantage of the learnable edge reweighting over heuristic-based graph rewriting lies in its **(1) end-to-end optimization** and **(2) structural preservation**. Specifically:
>
> - Unlike heuristic methods that rely on rigid rules (e.g., fixed similarity thresholds) as a pre-processing step, our masking spoiler is differentiable and jointly optimized with the anomaly detection objective. This allows the model to learn task-specific edge weights that maximize the discrimination between normal and anomalous clusters.
>
> - Furthermore, while heuristic-based methods ("hard" reweighting) perform binary removal that irreversibly destroys structural connections, our method is a "soft" reweighting strategy (continuous values in $[0,1]$) that selectively suppresses heterogeneous noise without severing message-passing paths. This is empirically supported by the ablation study (Table 2), which demonstrates the importance of the masking module. Additionally, Figure 5 also provides another empirical evidence that our model successfully learns to assign lower weights to inter-class edges while preserving intra-class connectivity.
>
> We will include these discussions in the revised version of our paper.
>
> **Response to Q3:** Thank you for your concern. We would like to answer your question via the following three aspects:
>
> - As described in Algorithm 1, we initialize the cluster centroids via $K$-Means on the initial node embeddings. Even before training, the GCN backbone serves as a Laplacian smoothing filter to aggregate local neighborhood attributes. Consequently, the initial embeddings $Z$ are not random noise, instead, they already capture the inherent structural and attribute patterns of the graph. This ensures that the initial pseudo-labels, while coarse, provide a statistically valid starting point rather than pure noise.
>
> - The optimization does not rely solely on the pseudo-labels. The reconstruction loss also serves as an essential regularizer, especially in the early phase. By forcing the latent embeddings to reconstruct the original node attributes and graph structure, it ensures the learned latent embeddings preserve the semantic information and prevent the encoder from overfitting to incorrect pseudo-labels or drifting into arbitrary manifolds.
>
> - To explicitly avoid converging to a trivial solution (a common failure mode where all nodes collapse into a single cluster), we impose a diversity loss to penalize the model if the sample proportion of any cluster falls below a threshold, which prevents the optimization from the "cluster collapse" issue and improves the stability of training.
>
>
> **We hope that the explanation and further analysis help to address your concerns.**

---

> > ### Comment · Reviewer_A5bT · 2025-11-24
> >
> > I would like to thank the authors for their rebuttal. The added experimental results and analyses have effectively addressed my concerns, particularly through the inclusion of more baselines and deeper analysis. Overall, the paper presents a solid solution to the "homophily trap" in graph anomaly detection. I am willing to increase my confidence to support this work.

---

> > > ### Author Response · Authors · 2025-11-25
> > >
> > > Thank you for your positive feedback. We are glad to hear that our rebuttal addressed your concerns.

---

### Comment · Area_Chair_2h28 · 2025-11-26

Dear reviewers,

The authors have provided updates and clarifications during the rebuttal, and they are awaiting your follow-up comments.

Please take a moment to review the updates if you haven't already.

Thank you for your efforts.

---

### Author Response · Authors · 2025-12-04
**Global Response to ACs, SACs, and PCs (1/3)**

Dear ACs/SACs/PCs,

We sincerely thank ACs/SACs/PCs for their time and efforts in the review process, and we appreciate all the reviewers for providing constructive comments on our paper. During the rebuttal phase, we made substantial improvements to address reviewers' concerns. **Although the initial scores (8, 4, 2, 2) of our paper are not satisfactory, following our active discussions with all reviewers, three reviewers (Reviewer A5bT, ErB5, and Q1xe) have explicitly confirmed that their concerns have been fully addressed and are willing to provide a positive rating for our paper (8, 6, 6)**. Reviewer 6mWp is also **willing to increase the rating** after providing new results on more datasets, which we have supplemented in the end. We would like to take this opportunity to **summarize (i) our discussions with all reviewers during the rebuttal and (ii) the key improvements and clarifications we have made, to facilitate the review process**.

---


### **(i) Overall Evaluation by Reviewers**
#### **Reviewer A5bT**
Reviewer A5bT initially gave our paper a **positive** rating and raised a few insightful concerns regarding (1) the choice of Chebyshev distance for MMD kernel computation, (2) additional parameter sensitivity analysis, (3) additional graph rewriting baseline comparison, (4) advantages of our method vs. heuristic-based edge editing, and (5) stability concern. After the rebuttal, Reviewer A5bT explicitly stated that the added experimental results and analyzes have **effectively addressed the concerns**, and our paper "**presents a solid solution to the 'homophily trap'** in graph outlier detection". **Reviewer A5bT is willing to increase confidence in supporting our work**.

---

#### **Reviewer ErB5**
Reviewer ErB5's initial concerns focused on (1) the clarity of concepts (e.g., $k$-hops away from anomalies), (2) the experiment on large-scale datasets, (3) the interaction between clustering and reconstruction, (4) the concern regarding the number of clusters, and (5) the follow-up concern regarding the practicality of hyperparameter selection. After our detailed responses and additional experiments, **Reviewer ErB5 stated that our responses had addressed all the concerns and is willing to raise the score to a positive rating**.

---

#### **Reviewer Q1xe**
Reviewer Q1xe initially raised concerns regarding (1) the novelty (our method vs. GAT/DEC), (2) the role of Proposition 1, (3) sensitivity to $k$-means initialization, (4) scalability to large graphs, and (4) empirical evidence to justify the learned weights. After our further clarifications (particularly regarding novelty), additional experiments on large graphs, and empirical analysis of learned weights in the rebuttal, **Reviewer Q1xe stated that all concerns have been addressed and raised the score to positive (from 2 to 6)**.

---

#### **Reviewer 6mWp**
Reviewer 6mWp's concerns centered on (1) the empirical complexity comparison, (2) the experiment on more GAD benchmarks, (3) comparison with more baselines, and (4) hyperparameter selection. In response, we added (1) running-time and memory comparisons, (2) evaluations on more GAD benchmarks (Reddit, OGB-Proteins, and Amazon), (3) more SOTA baselines (TAM, SmoothGNN, GADAM) suggested by Reviewer 6mWp, and (4) detailed clarifications on our hyperparameter selection strategy. In the final feedback, **Reviewer 6mWp acknowledges our efforts and is willing to increase the score after we demonstrated satisfactory real-world performance.** We have since added the requested Amazon results, where CER-GOD remains competitive against comprehensive SOTA baselines. Together with the datasets we added previously (OGB-Proteins and Reddit), we believe the added experiment has addressed this remaining point.

---

---

> ### Author Response · Authors · 2025-12-04
> **Global Response to ACs, SACs, and PCs (2/3)**
>
> ### **(ii) Key Improvements and Clarifications**
>
> #### **Strengthened Empirical Evaluation**
>
> - **(a) Additional SOTA baselines (Reviewers A5bT's W3, and 6mWp's W3):** We broadened the baseline set to include the latest GAD methods: **TAM[NeurIPS'23]**, **GADAM[ICLR'24]**, and **SmoothGNN[WWW'25]**. The experimental results clearly show that CER-GOD achieves competitive performance across all datasets.
>
> - **(b) Additional real-world and large-scale datasets (Reviewers ErB5's W2, Q1xe's Q3 and 6mWp's W2 \& FQ1):** We supplemented experiments on more datasets, including **Reddit (with 366 real-world outliers)**, **OGB-Proteins (132,534 nodes, 79M+ edges)**, and **Amazon (explicitly requested by Reviewer 6mWp)**, where the datasets are from the literature raised by Reviewer 6mWp. These additional experiments fully demonstrate the effectiveness and scalability of CER-GOD across the recent large-scale, real-world datasets.
>
> - **(c) Time and memory efficiency (Reviewer 6mWp's W1 \& FQ2):** We provided **running-time comparisons** on multiple datasets against several SOTA baselines, where the results demonstrated CER-GOD's efficiency. Reviewer 6mWp (FQ2) further raised concerns about memory usage. Therefore, we further provide a **memory-usage comparison** (per epoch) against baseline methods. The results also indicated CER-GOD’s memory consumption is competitive and close to the lighter baselines.
>
> ---
>
> #### **Methodological Clarifications and Ablations**
> - **(a) Distribution repulsion layer choice and diversity loss (Reviewer A5bT's W2 \& Q1):** We added layer-wise sensitivity experiments showing and explaining why applying distribution repulsion on the first encoder layer yields the best performance and that deeper layers gradually degrade performance. For the diversity loss hyperparameter $\epsilon$, we added an **extensive sensitivity analysis** demonstrating that CER-GOD is **robust across a reasonable range**, and only fails when $\epsilon$ is extremely small or too rigidly balanced.
> - **(b) Training stability and $k$-Means initialization (Reviewer A5bT's Q3 and Q1xe's Q2)**: We explicitly explain that: (1) $k$-Means is used only to initialize cluster centroids on latent embeddings; (2) centroids are adaptively updated during training; (3) reconstruction and diversity losses act as strong regularizers; and (4) we empirically verify robustness via **low variance across 10 runs with random $k$-means initialization**.
> - **(c) Reconstruction vs. Pure Clustering (Reviewer ErB5's W3):** We added a new **"w/o Reconstruction"** variant in the ablation study, where the learnable clustering is applied directly on latent embeddings. This variant performs significantly worse than the full model, which confirms that attribute and structure reconstruction are **crucial to preserve semantic information and topology**. We also clarified that the **previously reported "w/o SD-MS" variant already corresponds to "clustering without masking"**.
> - **(d) Number of Clusters and Multi-Modal Normals (Reviewer ErB5's W4 \& W5):** We clarified that the framework is **theoretically capable** of handling $c\geq 2$ clusters, but that in fully unsupervised anomaly detection, mapping multiple clusters to binary labels **requires unstable heuristics**. Following the setting in recent literature, we thus use the $c=2$ and provide the observed results varying $c\in \\{2,\dots,5\\}$, which show **consistent degradation** when increasing $c$. We also explain how the masking spoiler tends to align multi-modal normal patterns into a compact latent manifold while separating anomalies, which is validated by embedding visualization in our paper.
>
> ---

---

> ### Author Response · Authors · 2025-12-04
> **Global Response to ACs, SACs, and PCs (3/3)**
>
> #### **Novelty and Theoretical Insight**
> - **Our Masking Spoiler vs. Local Attention (Reviewer Q1xe's W1 \& Q1):** We mathematically distinguished our learnable mask $M\in \mathbb{R}^{N\times N}$, which is optimized via a global distribution repulsion loss, from: (1) GAT’s local feature-based attention coefficients; and (2) ADA-GAD’s smoothness-based relation-strength. The mathematical analysis validated that CER-GOD suppresses edges specifically because they reduce global cluster separability rather than because of local dissimilarity. This is crucial under the homophily trap, where anomalies mimic their neighbors.
> - **Self-Discriminative Module vs. DEC (Reviewer Q1xe's W2):** We clarified that DEC acts only in the feature space and does not change the topology. In contrast, our framework forms a closed feedback loop between the clustering-based detector and the learned mask from the masking spoiler. This loop allows CER-GOD to actively purify its own input graph structure, which DEC cannot.
> - **Proposition 1 and Empirical Evidence for Homophily Trap (Reviewer Q1xe's W3 \& Q4):** We clarified that the aim of Proposition 1 is not to provide a new bound, but to formally link the principles of information propagation to the context of the "Homophily Trap". We extended the homophily trap study to two other real-world datasets (Cora, Flickr), showing that 1-hop normal nodes exhibit shifts closer to anomalies than multi-hop normals. We additionally computed average mask weights on Flickr, showing a strong correlation between learned edge weights and homophily (high weights for homophilous edge pairs, low for heterophilous edge pairs), which provides quantitative evidence that the learned edge weights actually correlate with homophily. After these additions, Reviewer Q1xe explicitly stated that all concerns have been addressed.
>
> ---
>
> #### **Hyperparameters Selection**
> Reviewers 6mWp (W4 \& W5 \& FQ3) and ErB5 (FQ) misunderstood an important point about grid search in an unsupervised setting and the practicality of selecting hyperparameters on new datasets. To address these concerns,
> - (a) We explicitly clarified the **unsupervised protocol** used in the hyperparameters selection: for each dataset, we (1) fix a finite grid around default values (e.g., a logarithmic grid for $\alpha$ and $\beta$); (2) train each candidate for a fixed number of epochs; (3) select the configuration only using training-time quantities including final training loss and a simple non-collapse check on cluster assignments (to avoid trivial all-in-one-cluster solutions). Hence, the grid search is not used for training, but rather for **robustness checks** obviously, and we emphasize that **no labels are used in this hyperparameter selection process to avoid label leakage**.
> - (b) To facilitate the real-world development, we provided **recommended default ranges and explicit optimal values of each hyperparameter**. We also clarified that our parameter sensitivity analysis results showed that within a broad range (e.g., $\alpha$, $\beta$ $\in [0.01, 1]$), the performance of CER-GOD remains superior and better than strong SOTA baselines. This demonstrates that CER-GOD **does not require fine-grained tuning and can operate reliably with default settings on new data**.
>
> After our clarifications, Reviewer ErB5 and 6mWp confirm their concerns have been addressed and view the remaining suggestions as refinements to be documented in the appendix, rather than fundamental concerns.
>
> ---
>
> #### **Other Clarity and Presentation Improvements**
> We have substantially improved the clarity and consistency of the paper, which directly addresses the clarity issues highlighted by Reviewers ErB5 and A5bT:
> - (a) Added precise definitions of "1-hop away" and "multi-hop away" from anomalies.
> - (b) Corrected the "Data Distribution" module to "Latent MMD Score Distribution" and explained it more clearly.
> - (c) Updated Figure 2 with a legend, explicit pseudo-labeling flow, and consistent notation of $\mathbf{z}_{i}$.
> - (d) Provided empirical justification for using a Chebyshev-based kernel (outperforming RBF across multiple datasets), and added sensitivity analyses for the diversity loss parameter $\epsilon$ and the layer used in the MMD loss.
>
> ---
>
> **Given that all reviewers recognized the innovation and contribution of our work to the graph outlier detection community, we kindly request that the panel consider the above clarifications during the review process. We trust that, as one of the world’s leading CS conferences, the ICLR committee will provide a thorough and fair evaluation of the merit of our work.**
>
> Best Regards,
>
> Authors

---

### Meta-Review · Area_Chair_FVBC · 2026-01-06

**Summary:**

This paper studies graph outlier detection under the homophily trap.
Reviewers raised concerns about novelty, clarity, scalability, stability, and hyperparameter selection.
Some reviewers initially questioned whether the method was incremental and whether clustering-guided masking was sufficiently justified.
There were also concerns about experiments on large graphs, sensitivity to initialization, and practical use in unsupervised settings.
The rebuttal provided substantial new experiments, clearer theoretical distinctions, and detailed methodological clarifications. Most reviewers raised or agreed to raise their scores.

**Reviewer Concerns:**

Concerns on novelty, training stability and K-means initialisation, scalability, choice of kernel and diversity loss, and number of clusters are addressed during the rebuttal.

Concerns partially outstanding (minor):
- Some novelty concerns remain subjective but are largely mitigated by analysis and experiments.
- Hyperparameter selection still requires careful documentation, but practical guidance is now provided.

**Reviewer Scores:**

Reviewer A5bT raised score from 6 to 8.
Reviewer ErB5 agree to raised the initial score of 4, likely to 6.
Reviewers Q1xe and 6mWp explicitly mentioned that they are happy with the rebuttal. They are likely to improve their scores to 6.

---

### Decision · Program_Chairs · 2026-01-26

Accept (Poster)